EMBO
reports

# Lamin A-mediated nuclear lamina integrity is required for proper ciliogenesis

Jia-Rong Fan[1,2] [iD], Li-Ru You[1,2], Won-Jing Wang[1], Wei-Syun Huang[1] [iD], Ching-Tung Chu[1], Ya-Hui Chi[3] [iD] & Hong-Chen Chen[1,2,*] [iD]

## Abstract

**The primary cilium is a sensory organelle that receives specific signals from the extracellular environment important for vertebrate development and tissue homeostasis. Lamins, the major components of the nuclear lamina, are required to maintain the nuclear structure and are involved in most nuclear activities. In this study, we show that deficiency in lamin A/C causes defective ciliogenesis, accompanied by increased cytoplasmic accumulation of actin monomers and increased formation of actin filaments. Disruption of actin filaments by cytochalasin D rescues the defective ciliogenesis in lamin A/C-depleted cells. Moreover, lamin A/C-deficient cells display lower levels of nesprin 2 and defects in recruiting Arp2, myosin Va, and tau tubulin kinase 2 to the basal body during ciliogenesis. Collectively, our results uncover a functional link between nuclear lamina integrity and ciliogenesis and implicate the malfunction of primary cilia in the pathogenesis of laminopathy.**

**Keywords** cilia; ciliogenesis; ciliopathy; lamin; laminopathy
**Subject Categories** Cell Adhesion, Polarity & Cytoskeleton

## Introduction

The primary cilium, an antenna-like organelle that harbors specific signaling receptors, is required for key developmental processes such as the Hedgehog signaling (Singla & Reiter, 2006). Defects in cilia biogenesis (*i.e.,* ciliogenesis) cause numerous diseases, collectively termed ciliopathies (Fliegauf *et al*, 2007). The patients with ciliopathies often present with multi-system pathologies (Forsythe & Beales, 2013), such as cystic kidney, retinal degeneration, hypogonadism, situs in versus, and polydactyly. Ciliogenesis is a well-orchestrated process that is initiated upon cells enter G0 (quiescence) or G1 phase (Sanchez & Dynlacht, 2016). The core structure of the cilium is a membrane-sheathed, microtubule axoneme that

extends from a basal body, which is derived from the mother centriole. Three key steps are essential for converting a mother centriole into a basal body. First, at the distal appendage (DA) of the mother centriole, small preciliary vesicles (PCVs) accumulate and fuse into a larger ciliary vesicle (CV). This initial stage is facilitated by many molecules, such as CEP164 and Rab11 (to recruit PCVs), myosin Va (to transport PCVs), and EHD1 (to fuse PCVs into CV) (Westlake *et al*, 2011; Schmidt *et al*, 2012; Lu *et al*, 2015; Wu *et al*, 2018). Second, tau tubulin kinase 2 (TTBK2) is recruited by CEP164 to the distal end of the mother centriole, which is required for the onset of the third key step—the removal of CP110, a repressor of cilia (Spektor *et al*, 2007; Cajanek & Nigg, 2014; Liao *et al*, 2015). Before ciliation, CP110 localizes to the distal ends of both mother and daughter centrioles, where it forms a "cap" that restrains microtubule growth. Once the ciliogenic program is triggered, CP110 is asymmetrically removed from the TTBK2-enriched mother centriole (Goetz *et al*, 2012). This final step releases the inhibition of microtubule growth and then axonemal elongation begins.

Although cilia are microtubule-based structures, recent studies reveal that ciliogenesis is also modulated by other types of the cytoskeleton, such as the actin filaments (May-Simera & Kelley, 2012). The actin filaments emerge as a key suppressor of ciliogenesis (Yan & Zhu, 2012). Generally, globular actin monomers (G-actin) spontaneously polymerize into filaments (F-actin), which can be promoted or severed by various accessory proteins (Pollard & Cooper, 2009). Loss of cortactin or Arp3, both of which are the F-actin-promoting proteins, increases ciliogenesis (Bershteyn *et al*, 2010; Kim *et al*, 2010). On the contrary, depletion of the F-actin-severing protein gelsolin or cofilin decreases ciliogenesis (Kim *et al*, 2010, 2015). Disruption of F-actin by cytochalasin D facilitates ciliogenesis and elongates the length of cilia (Bershteyn *et al*, 2010; Kim *et al*, 2010; Cao *et al*, 2012). Moreover, the microRNA mir-129-3p promotes ciliogenesis by concomitantly down-regulating four F-actin promoting proteins, ABLIM1, ABLIM3, TOCA1, and Arp2 (Cao *et al*, 2012). In addition to the microfilament, the microtubule is involved in transporting ciliary vesicles and pericentriolar matrix proteins to the basal body during ciliogenesis (Kim *et al*, 2008; Westlake *et al*, 2011). However, the role of the intermediate filaments in ciliogenesis remains unexplored.

1   Institute of Biochemistry and Molecular Biology, National Yang-Ming University, Taipei, Taiwan
2   Cancer Progression Research Center, National Yang-Ming University, Taipei, Taiwan
3   Institute of Biotechnology and Pharmaceutical Research, National Health Research Institutes, Zhunan, Taiwan
    *Corresponding author. Tel: +886 2 28267123; Fax: +886 2 28201886; E-mail: hcchen1029@ym.edu.tw

Lamins that represent the type V intermediate filaments are the major component of the nucleoskeleton (Broers *et al*, 2006; Simon & Wilson, 2011). They not only maintain the structure of the nucleus, but also participate in various nuclear activities, such as DNA replication, transcription, and DNA repair, through their interaction with chromatins, transcriptional factors, and signaling molecules (Spann *et al*, 2002; Ivorra *et al*, 2006; Nitta *et al*, 2006; Gonzalez-Suarez *et al*, 2009). More than 400 mutations (the UMD-LMNA mutation database at http://www.umd.be/LMNA/) in *Lmna* (the gene encoding lamin A/C) have been attributed to at least 11 diseases, which are collectively termed laminopathies (Capell & Collins, 2006). The most striking laminopathy is Hutchinson-Gilford progeria syndrome (HGPS), which exhibits premature aging phenotypes including atherosclerosis, loss of subcutaneous fat, alopecia, and early death in teens (Hutchinson, 1886; Kudlow *et al*, 2007). HGPS results from a dominant-negative form of a farnesylated lamin A protein called progerin (Eriksson *et al*, 2003). The expression of progerin causes the deformation of nuclear envelope, clustering of nuclear pores, defective signaling pathways, and activation of senescence program (Goldman *et al*, 2004; Liu *et al*, 2005; Varela *et al*, 2005). Another common laminopathy is Emery-Dreifuss muscular dystrophy (EDMD), which is characterized by muscle-specific phenotypes, including joint contractures, muscle degeneration, and cardiomyopathy (Emery, 1989). In spite of many prior studies, the pathogenesis for different laminopathies remains incompletely understood.

Although lamins form nuclear lamina in the nucleus, they play a crucial role in organizing the cytoskeleton by anchoring the LINC complexes (links the nucleoskeleton and cytoskeleton) (Simon & Wilson, 2011). Lamin A/C and emerin (a LINC protein) are involved in modulating the actin dynamics (Ho *et al*, 2013). Nesprin 2, a multi-isomeric protein of the LINC complexes, binds lamin A and emerin at the inner nuclear envelope and the cytoplasmic F-actin at the outer nuclear envelope (Zhang *et al*, 2002, 2005). Nesprin 2 depletion was reported to involve in ciliogenesis through actin remodeling (Dawe *et al*, 2009). In addition, the mice harboring a *Lmna* L52R mutation (Odgren *et al*, 2010) exhibit impaired cilia in their middle ears and hearing loss (Zhang *et al*, 2012). Interestingly, laminopathies and ciliopathies seem to share some common phenotypes, such as diabetes, craniofacial abnormalities, hearing loss, short stature, and abnormal fat and bone tissues. Whether ciliary abnormality is responsible for some of the phenotypes in laminopathies remains unclear. In this study, we show that the skin fibroblasts derived from HGPS patients display impaired ciliogenesis. The primary cilia lengths in many adult tissues from *Lmna* null mice are shorter than those of their littermate controls. The depletion of lamin A/C or nesprin 2 causes increased F-actin formation and defective ciliogenesis in human retinal pigment epithelial (RPE) cells. Our results support an important role of the nuclear lamina integrity in ciliogenesis and implicate that malfunction of primary cilia may be involved in the pathogenesis of laminopathies.

# Results

### Skin fibroblasts derived from HGPS patients show deficiency in ciliogenesis

To examine the potential impact of laminopathy on ciliogenesis, skin fibroblasts derived from five HGPS patients and two normal individuals were employed. The progerin expression and primary cilia in those cells were examined by immunofluorescence staining (Fig 1A). As expected, the fluorescence intensity of progerin in the HGPS fibroblasts was higher than that in the normal fibroblasts; however, the fluorescence intensity was quite variable among the HGPS fibroblasts (Fig 1A and B, Appendix Fig S1), which may be because the expression level of progerin is age-dependent and cumulative with cell passages in HGPS patient-derived fibroblasts (Goldman *et al*, 2004). The cell with the fluorescence intensity of progerin above $3.5 \times 10^6$ arbitrary units (A.U.) was defined as the one with a high-progerin level. The high-progerin HGPS fibroblasts had an enlarged, wrinkled nucleus and displayed deficiency in ciliogenesis (Fig 1 and Appendix Fig S1). Moreover, the high-progerin HGPS fibroblasts were more spread (Fig 1C and Appendix Fig S2A) and possessed more prominent actin filaments than the normal fibroblasts (Fig 1D and Appendix Fig S2B). Whereas detected in 90% of the normal fibroblasts, primary cilia were detected in ~50–60% of the high-progerin HGPS fibroblasts upon serum starvation (Fig 1E). It is known that slight deviation from a normal range in the length of cilia could affect their functions (Avasthi & Marshall, 2011). The length of primary cilia in the high-progerin HGPS fibroblasts was ~20–40% shorter than the normal fibroblasts (Fig 1F). Furthermore, the correlation analysis demonstrated that the progerin level was inversely correlated with the cilia length in all of the five HGPS patients' fibroblasts (Fig EV1). Using Arl13b, an axonemal marker, we confirmed that both cilia incidence and length were decreased in the high-progerin HGPS fibroblasts (Appendix Fig S3). Disruption of the F-actin by cytochalasin D restored the assembly and length of cilia in the high-progerin HGPS fibroblasts (Fig 1E and F). These data suggest that progerin may cause a defect in ciliogenesis through an actin-dependent manner.

### *Lmna* null mice display defective primary cilia in the skeletal muscle, kidney, uterus, ovary, and embryonic fibroblasts

To further examine the links between *Lmna* and ciliogenesis, cilia in various tissues from *Lmna* null mice and their littermate controls were stained with an acetylated tubulin antibody. *Lmna* null mice were developed and used as a model for EDMD (Sullivan *et al*, 1999). They showed reduced size in the tibialis anterior skeletal muscle, one of the characteristics of muscular dystrophy (Fig 2A). Interestingly, the cilia of the fibro/adipogenic progenitor cells in the skeletal muscle (Kopinke *et al*, 2017) of *Lmna* null mice were apparently shorter (1.66 ± 0.07 μm versus 2.72 ± 0.21 μm) and fewer than those of the control (Fig 2A, E and F). Next, we examined whether lamin A deficiency causes defects in the cilia in the kidney, the reproductive and respiratory systems, which are often affected in ciliopathy (Fliegauf *et al*, 2007; Forsythe & Beales, 2013). In the kidney, the cilia length of renal tubule epithelial cells in *Lmna* null mice was about 0.6 μm shorter than those in the control mice (Fig 2B and E). The cilia of the stromal cells in the uterus were remarkably shorter (1.37 ± 0.03 μm versus 2.44 ± 0.12 μm) and fewer in *Lmna* null mice (Fig 2C, E and F, Appendix Fig S4). Also, the cilia of the granulosa cells in the follicles of the ovary were significantly shorter in *Lmna* null mice (Fig 2D and E, Appendix Fig S4). However, the motile cilia lining the oviducts and bronchi appeared normal in *Lmna* null mice compared with their controls

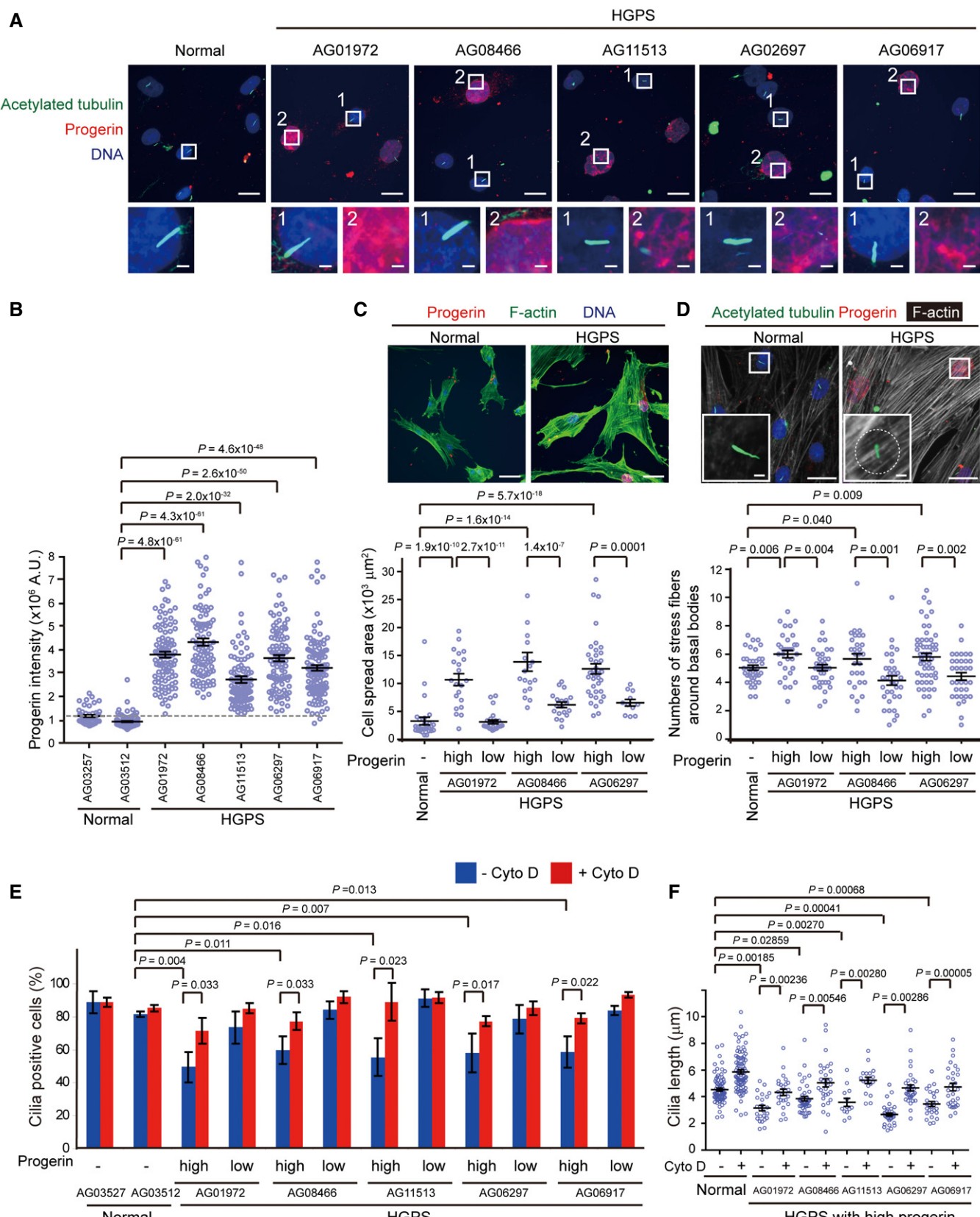

Figure 1.

**Figure 1. Skin fibroblasts derived from HGPS patients display deficiency in ciliogenesis.**

A Normal human fibroblasts and HGPS fibroblasts were serum-starved for 48 h and stained for progerin (red), acetylated tubulin (green), and DNA (blue). The representative images show that the cell with high progerin (inset 2), but not low progerin (inset 1), displays defective cilia formation. Scale bars, 20 or 2 μm (in the magnified images).

B The progerin fluorescence intensity of the cells was measured by the Zeiss ZEN2 software ($n \geq 148$). The dashed line indicates the background level of progerin. A.U., arbitrary units.

C The cells were grown in the growth medium and stained for progerin (red) and F-actin with phalloidin (green). The cell with high or low progerin expression was defined by the progerin fluorescence intensity above or below $3.5 \times 10^6$ (A.U.), respectively. The cell spreading area was measured according to the phalloidin staining ($n \geq 30$). Scale bars, 50 μm.

D The cells were serum-starved for 32 h to allow cilia formation and then treated with (+) or without (−) 250 nM cytochalasin D (Cyto D) for another 16 h before fixation. The cells were stained for progerin (red), acetylated tubulin (green), F-actin (white), and DNA (blue). The numbers of actin filaments within 60 μm$^2$ around the cilia (as illustrated by the circle) from the cells under serum starvation without Cyto D treatment were measured ($n \geq 55$). Scale bars, 20 or 2 μm (in the magnified insets).

E, F The cells were subjected to immunofluorescence stain as in (D). The percentage of the cells with cilia in the total counted cells (E, $n \geq 50$) and the cilia length (F, $n \geq 50$) were measured.

Data information: In B, C, D and F, values (means ± SEM) are from three independent experiments. In E, values (means ± SD) are from three independent experiments. Statistical significance of differences is assessed with a Student's *t*-test.

(Appendix Fig S5). These data suggest that lamin A/C may play a pivotal role in the formation of primary cilia but not motile cilia.

**Impaired ciliogenesis by the depletion of lamin A/C may partially result from nesprin 2 suppression**

To examine the necessity and the underlying mechanism of lamin A/C in ciliogenesis, lamin A/C in RPE cells was depleted by short-hairpin RNA (shRNA) specific to lamin A or luciferase as a control (Fig 3A). The depletion of lamin A/C led to abnormal nuclear shape and loss of the nuclear integrity, as manifested by the cytoplasmic distribution of emerin (Appendix Fig S6). More importantly, the depletion of lamin A/C significantly decreased the number and length of primary cilia (Fig 3B–D and Appendix Fig S7), which was restored by re-expression of lamin A (Fig 3B–D and Appendix Fig S7). Furthermore, the cilia incidence over time after serum withdrawal was monitored. The result supported that the deficiency of lamin A/C led to attenuated cilia formation (Fig 3E). To further confirm the results, lamin A/C was knocked out in RPE cells through CRISPR/Cas9 gene editing technology (Fig EV2A). Consistently, the lamin A/C-knockout led to abnormal cytoplasmic distribution of emerin (Fig EV2B) and defective ciliogenesis (Fig EV2C and D). These data indicate that lamin A/C is required for ciliogenesis.

The lamin A/C-depleted RPE cells exhibited enlarged cell morphology (Fig 4A and B), accompanied by increased formation of actin filaments (Fig 4A) and focal adhesions (Fig 4A and C). Most of these actin filaments were likely stress fibers because they traversed the cell and terminated at paxillin-labeled focal adhesions (Fig 4A and C). As the result, more stress fibers were found to surround the basal bodies of the lamin A/C-depleted cells (Fig 4D and E). Disruption of the actin filaments by cytochalasin D significantly restored the number and length of cilia (Fig 4D, F and G). Accordingly, mouse embryonic fibroblasts (MEFs) derived from *Lmna* null mice showed increased cell spreading (Appendix Fig S8A), more prominent F-actin (Appendix Fig S8B), and decreased cilia incidence and length, all of which were restored by cytochalasin D treatment (Appendix Fig S8C–E). These data suggest that deficiency in lamin A/C may somehow increase the formation of stress fibers, which has an adverse effect on cilia formation. To evaluate whether cilia-mediated Sonic Hedgehog (Shh) signaling

pathway is defective in lamin A-depleted cells, the MEFs ($Lmna^{+/+}$ versus $Lmna^{-/-}$) were treated with the Shh-agonist SAG (Chen *et al*, 2002) to activate the Shh signaling. We found that the Shh signaling appeared not affected in $Lmna^{-/-}$ MEFs by measuring the transcription of Gli1 (Appendix Fig S9).

Nesprin 2, a fundamental unit of the LINC complex, connects to lamin A at the inner nuclear envelope and cytoplasmic F-actin at the outer nuclear envelope (Zhen *et al*, 2002). We unexpectedly found that depletion of lamin A/C apparently decreased the level of nesprin 2 in RPE cells (Fig 5A) and $Lmna^{-/-}$ MEFs (Appendix Fig S10). This raises a possibility that the effect of lamin A/C depletion on actin filaments and cilia may be at least partially through nesprin 2 suppression. Indeed, similar to lamin A/C depletion, nesprin 2 depletion also led to increased stress fibers (Fig 5B–D), decreased cilia formation (Fig 5E), and reduced cilia length (Fig 5F). Again, disruption of the actin filaments by cytochalasin D restored the number and length of cilia in nesprin 2-depleted cells (Fig 5E and F). These results suggest that lamin A/C and nesprin 2 may regulate ciliogenesis by modulating the actin homeostasis.

**Deficiency in ciliogenesis by the depletion of lamin A/C or nesprin 2 may be caused by disruption of the actin homeostasis**

It is known that a large portion of the G-actin is stored as building blocks in the nucleus (Gonsior *et al*, 1999; Huff *et al*, 2004; Grosse & Vartiainen, 2013), and G-actin can shuttle between the nucleus and cytoplasm via nuclear pore complexes (Pendleton *et al*, 2003; Grosse & Vartiainen, 2013; de Lanerolle & Serebryannyy, 2015). Deoxyribonuclease I (DNase I), which binds to G-actin with a high affinity (Mannherz *et al*, 1980) has been used to detect G-actin in the cells (Huff *et al*, 2004). While 73.6% of the total G-actin was detected in the nucleus of the control RPE cells, ~50% of that was detected in the nucleus of the cells with a depletion of lamin A/C or nesprin 2 (Fig 6A and B). The subcellular fractionation also revealed that the relative ratio of nuclear actin in lamin A/C-depleted RPE cells was 32% lower than in control cells (Appendix Fig S11). Consistently, decreased distribution of G-actin in the nucleus was also observed in HGPS fibroblasts and $Lmna^{-/-}$ MEFs (Fig EV3). These results suggest that lamin A/C and nesprin 2 may function to

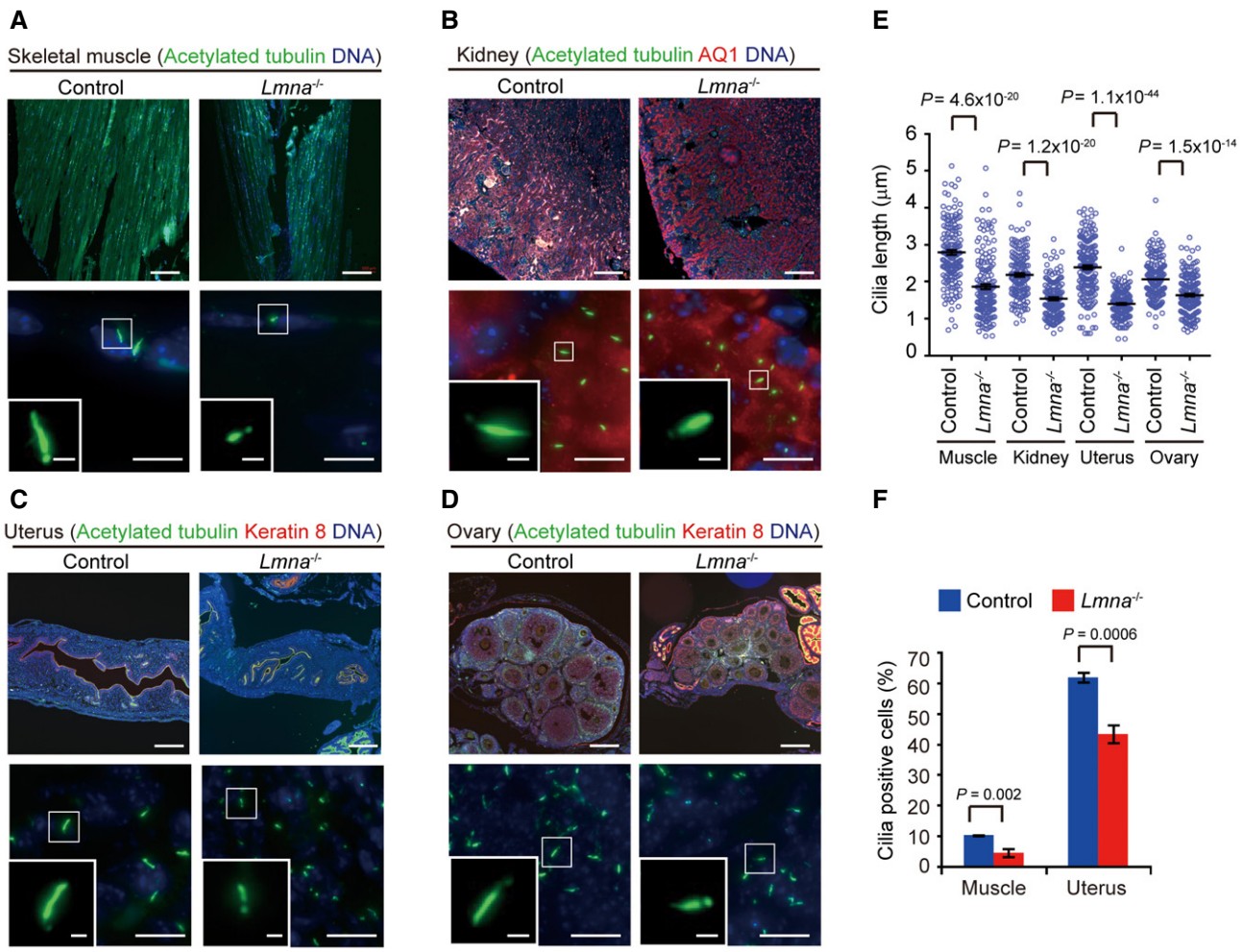

**Figure 2.** *Lmna* null mice display defective primary cilia in the skeletal muscle, kidney, uterus, and ovary.

A    The skeletal (tibialis anterior) muscles from control (*Lmna*[+/+] and *Lmna*[+/−]) and *Lmna*[−/−] mice were stained for acetylated tubulin (green) and nucleus (blue). The representative images with low and high magnification are shown. The insets are enlarged images for cilia.

B    The kidneys from control and *Lmna*[−/−] mice were stained for acetylated tubulin (green) and AQ1 (a marker for the proximal tubules, red). The insets show the cilia of proximal renal tubule cells.

C    The uterus from control and *Lmna*[−/−] mice was stained for acetylated tubulin (green) and keratin 8 (red). The insets show the cilia at the stroma of the uterus.

D    The ovary from control and *Lmna*[−/−] mice was stained for acetylated tubulin (green) and keratin 8 (red). The insets show the cilia of the granulosa cells in the follicles of the ovary.

E, F  The cilia length (E, n = 150–180) and cilia-positive cells (F, n ≥ 1197) in each organ from 4-week-old *Lmna*[−/−] mice (n = 3) and their littermate controls (n = 3) were measured.

Data information: Scale bars, 200 μm (A–D, upper panels); 10 μm (A–D, lower panels); 1 μm (A–D, insets). In E, values (means ± SEM) are from three independent experiments. In F, values (means ± SD) are from three independent experiments. Statistical significance of differences is assessed with Student's *t*-test.

preserve G-actin within the nucleus. Therefore, the deficiency in lamin A/C and/or nesprin 2 may lead to increased concentration of G-actin at the cytoplasm, which thereby facilitates the dynamic equilibrium toward F-actin assembly. To further examine this hypothesis, G-actin was arrested in the cytoplasm by depleting importin 9, which was reported to deliver G-actin into the nucleus (Dopie *et al*, 2012). Indeed, the depletion of importin 9 in RPE cells decreased the ratio of G-actin in the nucleus (Fig 6C–E), accompanied by increased cell spreading (Fig 6F). More importantly, it reduced the ciliogenesis (Fig 6G–I), which was restored by cytochalasin D (Fig 6H and I).

A reciprocal relationship between cilia and the cell cycle has been proposed (Quarmby & Parker, 2005). To examine whether the suppression of lamin A, nesprin 2 or importin 9 affects the cell cycle, the cell cycle of the RPE cells expressing shRNAs specific to those molecules was evaluated by flow cytometry. While HGPS fibroblasts showed about 10% increase in the G1 population (Appendix Fig S12A), the cell cycle of *Lmna*[−/−] MEFs or RPE cells with the shRNAs for suppressing lamin A, nesprin 2, or importin 9 was not affected (Appendix Fig S12B and C). Therefore, the cilia defects observed in this study less likely resulted from abnormal cell cycle.

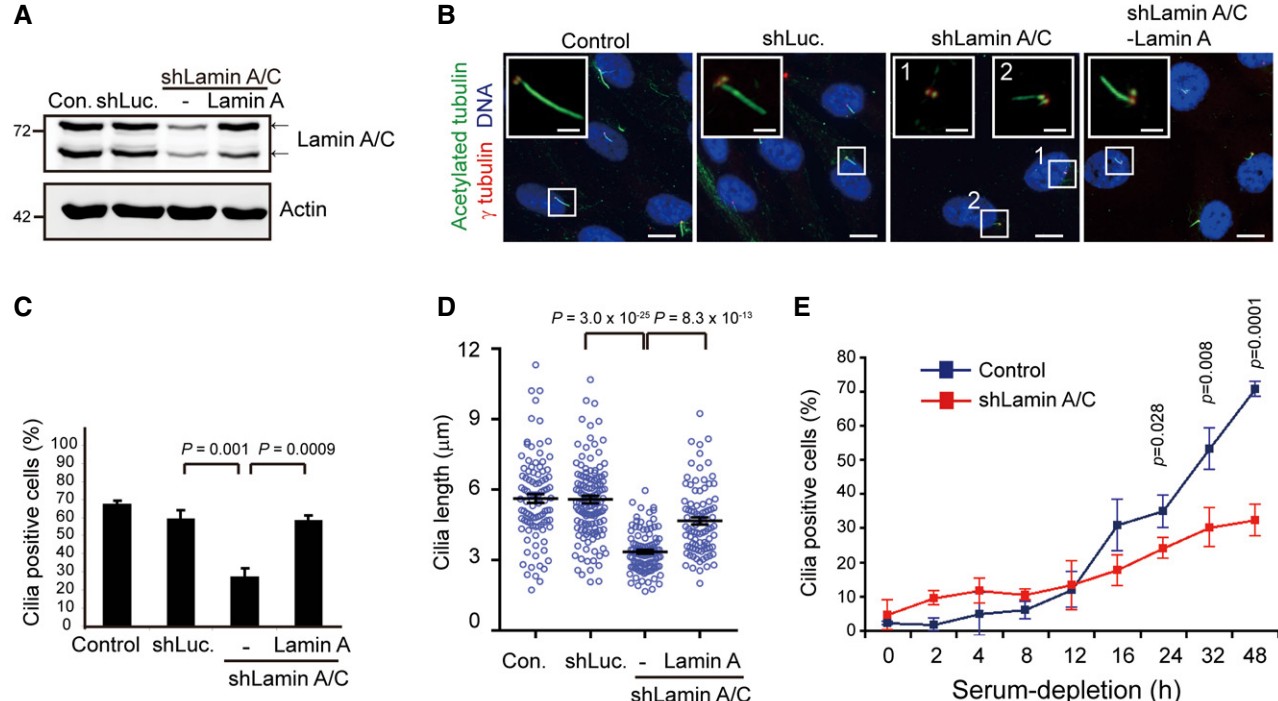

**Figure 3. Depletion of lamin A/C expression causes impaired ciliogenesis in RPE cells.**

A   RPE cells were infected with lentiviruses encoding shRNAs specific to lamin A/C (shLamin A/C) or luciferase (shLuc). Exogenous lamin A was re-expressed in the lamin A/C-depleted cells (shLamin A/C-Lamin A). An equal amount of whole-cell lysates was analyzed by immunoblotting with antibodies as indicated.

B   The cells were serum-starved for 48 h and stained for acetylated tubulin (green), γ-tubulin (red), and DNA (blue). The depletion of lamin A/C (shLamin A/C) led to impaired ciliogenesis, as manifested by no cilia (inset 1) or short cilia (inset 2). Scale bars, 20 or 2 μm (insets).

C   The percentage of the cells with cilia in the total counted cells ($n \geq 626$) was measured.

D   The length of cilia was measured ($n \geq 87$ in each group).

E   Cells were stained for acetylated tubulin (green) and γ-tubulin (red) after various intervals of serum starvation. The percentage of cilia-positive cells at each time point ($n \geq 372$) was measured.

Data information: In C and E, values (means ± SD) are from three independent experiments. In D, values (means ± SEM) are from three independent experiments. Statistical significance of differences is assessed with Student's *t*-test.

Source data are available online for this figure.

## Depletion of lamin A/C or nesprin 2 causes a defect in docking of Arp2 and myosin Va-mediated preciliary vesicles to the basal body

Recent studies show that myosin Va carries PCVs along the Arp2-mediated centrosomal actin network to the mother centriole to form CV (Wu *et al*, 2018). In this study, we surprisingly found that Arp2 localized exclusively at the mother centriole in the control RPE cells (Fig 7A and Appendix Fig S13A), but it was substantially detected at both mother and daughter centrioles in lamin A/C- or nesprin 2-depleted cells (Fig 7A). The centrosome with Arp2 at both mother and daughter centrioles failed to form cilium (Fig 7B and Appendix Fig S13B). F-actin promoting proteins Arp3 and WASP (Pollard & Cooper, 2009) were also found to localize at the centrosomes of ciliated RPE cells (Appendix Fig S14). However, unlike Arp2, they were not

**Figure 4. Depletion of lamin A/C increases the formation of F-actin, which has an adverse effect on ciliogenesis in RPE cells.**

A–C   The RPE cells as described in Fig 3 were stained for paxillin (as a marker for focal adhesion; green), F-actin (red), and DNA (blue). The representative images are shown (A). Scale bars, 20 or 5 μm (insets). The cell spreading area of the cells (B, $n \geq 108$) and the number of focal adhesions per cell (C, $n \geq 30$) were measured.

D–G   The cells were serum-starved for 32 h and then treated with (+) or without (−) Cyto D (250 nM) for another 16 h. The cells were stained for acetylated tubulin (white), pericentrin (green), F-actin (red), and DNA (blue). The representative images are shown (D). The left insets show the merged signals of acetylated tubulin and pericentrin. Scale bars, 20 or 2 μm (magnified images). The numbers of actin filaments within 60 μm² around the basal body (as illustrated by the circle) from the cells under serum starvation without Cyto D treatment were measured (E, $n \geq 132$). The percentage of the cells with cilia in the total counted cells (F, $n \geq 357$) and the length of cilia (G, $n \geq 126$) were measured.

Data information: In B, C, E and F, values (means ± SD) are from three independent experiments. In G, values (means ± SEM) are from three independent experiments. Statistical significance of differences is assessed with Student's *t*-test. For clarity, not all significances are indicated.

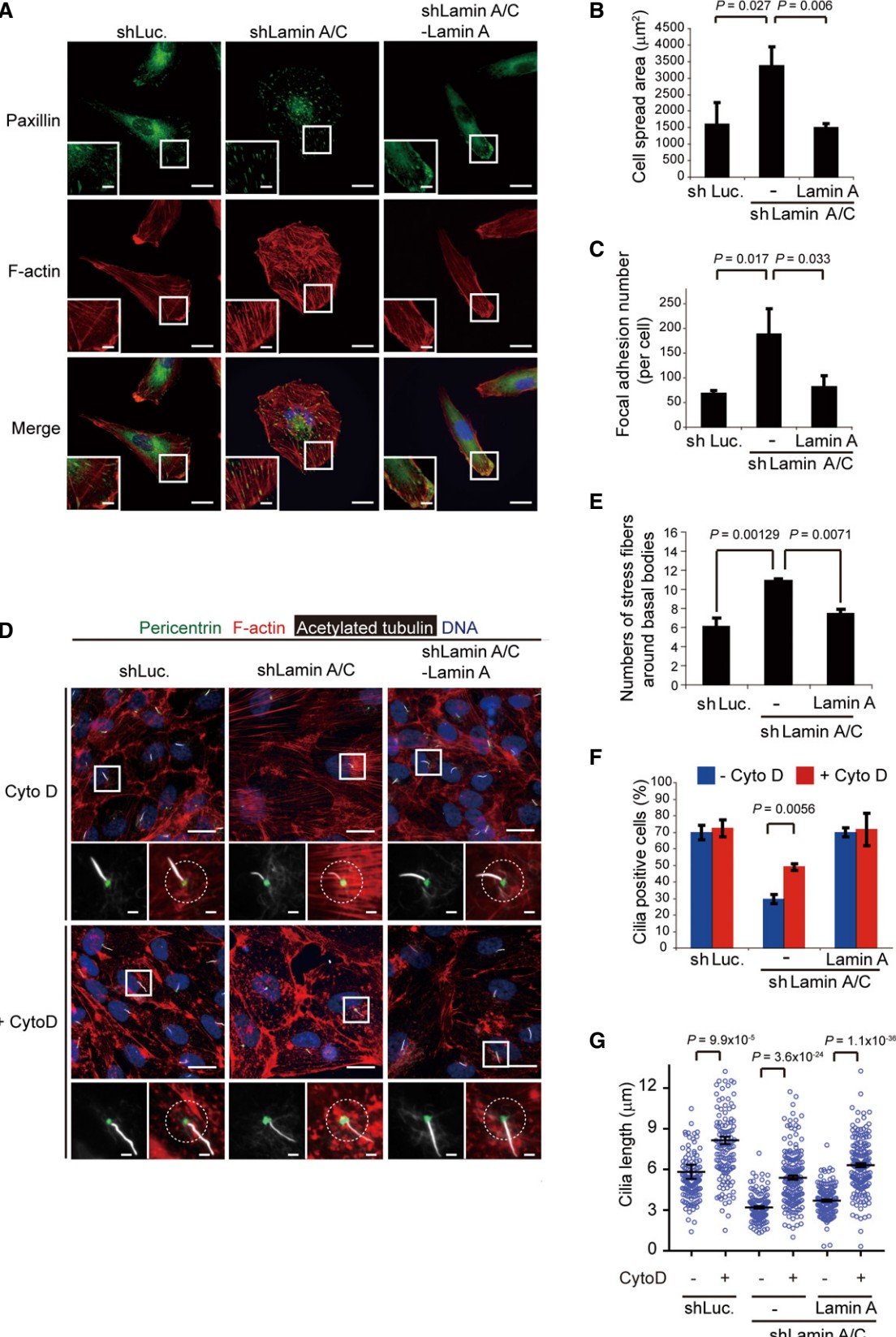

**Figure 4.**

localized exclusively at the basal bodies (Appendix Fig S14). Similar to Arp2, myosin Va was found to mainly localize only at the mother centriole in the control RPE cells (Fig 7C). Depletion of lamin A/C and nesprin 2 decreased this ratio to 26% and 36%, respectively (Fig 7C). The cells with the depletion of lamin A/C or nesprin 2 showed a diffusive staining of myosin Va around the mother and daughter centrioles (Fig 7C), implicating

that myosin Va-associated PCVs may accumulate around both centrioles but fail to convert into CV at the mother centriole. Consistently, high-progerin HGPS fibroblasts also showed defective docking of Arp2 (Fig EV4A) and myosin Va (Fig EV4B) to the basal body. These data suggest that the nuclear lamina integrity may be important for precise trafficking of Arp2 and myosin Va-PCV to the basal body during early ciliogenesis.

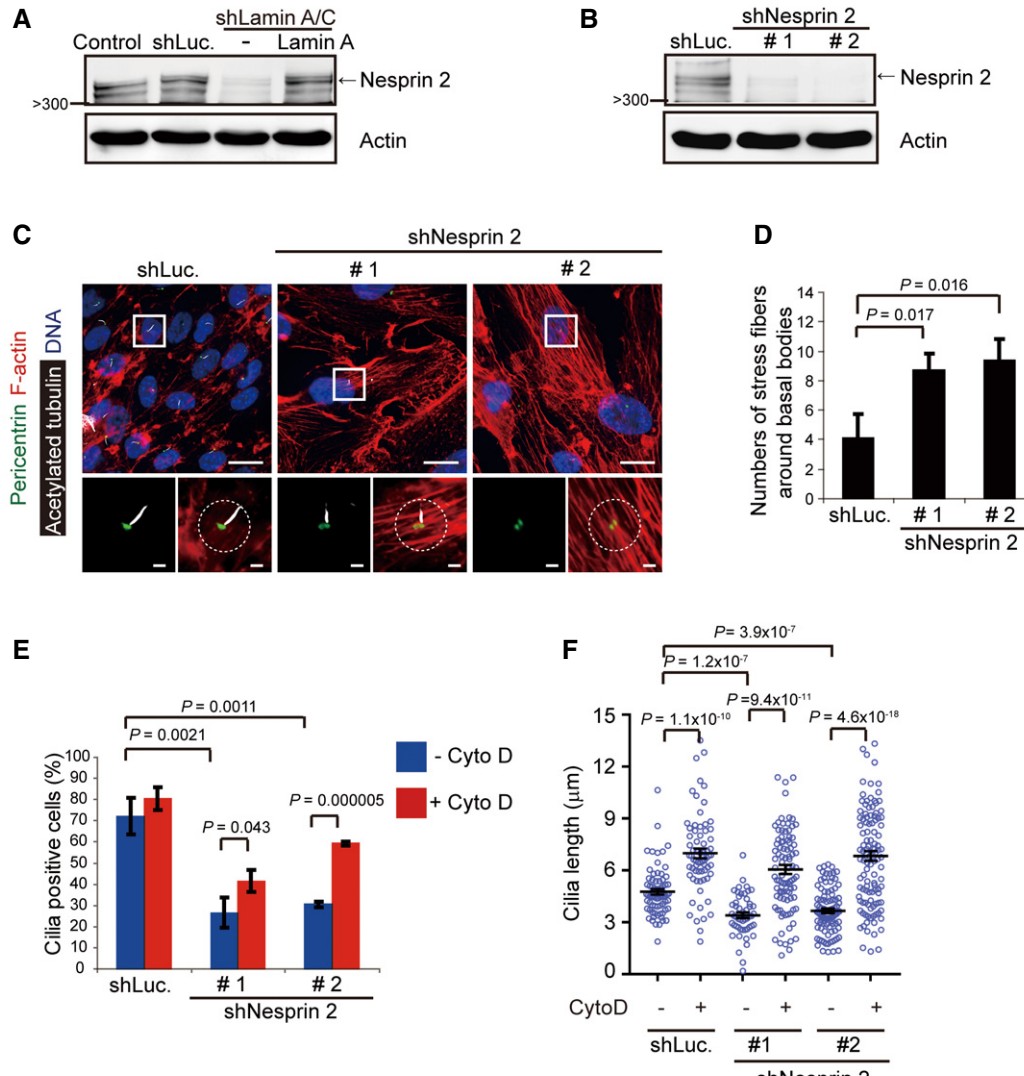

**Figure 5.  Impaired ciliogenesis by depletion of lamin A/C may partially result from nesprin 2 suppression.**

A      An equal amount of whole-cell lysates from RPE cells or those expressing shLamin A/C or luciferase (shLuc) was analyzed by the immunoblotting with antibodies as indicated.

B–D   RPE cells were infected with lentiviruses expressing shRNAs to nesprin 2 (shNesprine 2, clone #1, and clone #2) or luciferase (shLuc) as a control. An equal amount of whole-cell lysates was analyzed by immunoblotting with antibodies as indicated (B). The cells were serum-starved for 48 h and then stained for acetylated tubulin (white), pericentrin (green), F-actin (red), and DNA (blue). The representative images are shown (C) The left insets show the merged signals of acetylated tubulin and pericentrin. Scale bars, 20 or 2 μm (magnified images). The numbers of F-actin stress fibers within 60 μm$^2$ around the basal bodies (as illustrated by the circle) were measured (D, $n \geq 53$).

E, F   RPE cells expressing shNesprine 2 or shLuc were serum-starved for 32 h and then treated with (+) or without (−) Cyto D for another 16 h. The percentage of the cells with cilia in the total counted cells (E, $n \geq 197$) and the length of cilia (F, $n \geq 52$) were measured.

Data information: In D and E, values (means ± SD) are from three independent experiments. In F, values (means ± SEM) are from three independent experiments. Statistical significance of differences is assessed with Student's t-test.

## Depletion of lamin A/C suppresses the recruitment of TTBK2 to the mother centriole

It has been shown that the formation of CV requires complete DA of the basal body (Schmidt *et al*, 2012; Tanos *et al*, 2012). The depletion of the core DA component led to accumulated myosin Va-associated PCVs (Wu *et al*, 2018), similar to what we observed in lamin A/C-depleted RPE cells (Fig 7C). CEP164 is a key DA component of

the basal body, which is essential for TTBK2 recruitment and the following removal of CP110 at the basal body (Tsang & Dynlacht, 2013; Cajanek & Nigg, 2014). We found that CEP164 was normally localized to the mother centrioles of the control RPE cells and the lamin A/C-depleted cells, no matter whether they were ciliated or not (Fig 8A), indicating that lamin A/C depletion does not affect the DA maturation. As expected, TTBK2-recruitment and CP110-removal were observed at almost all basal bodies in ciliated cells

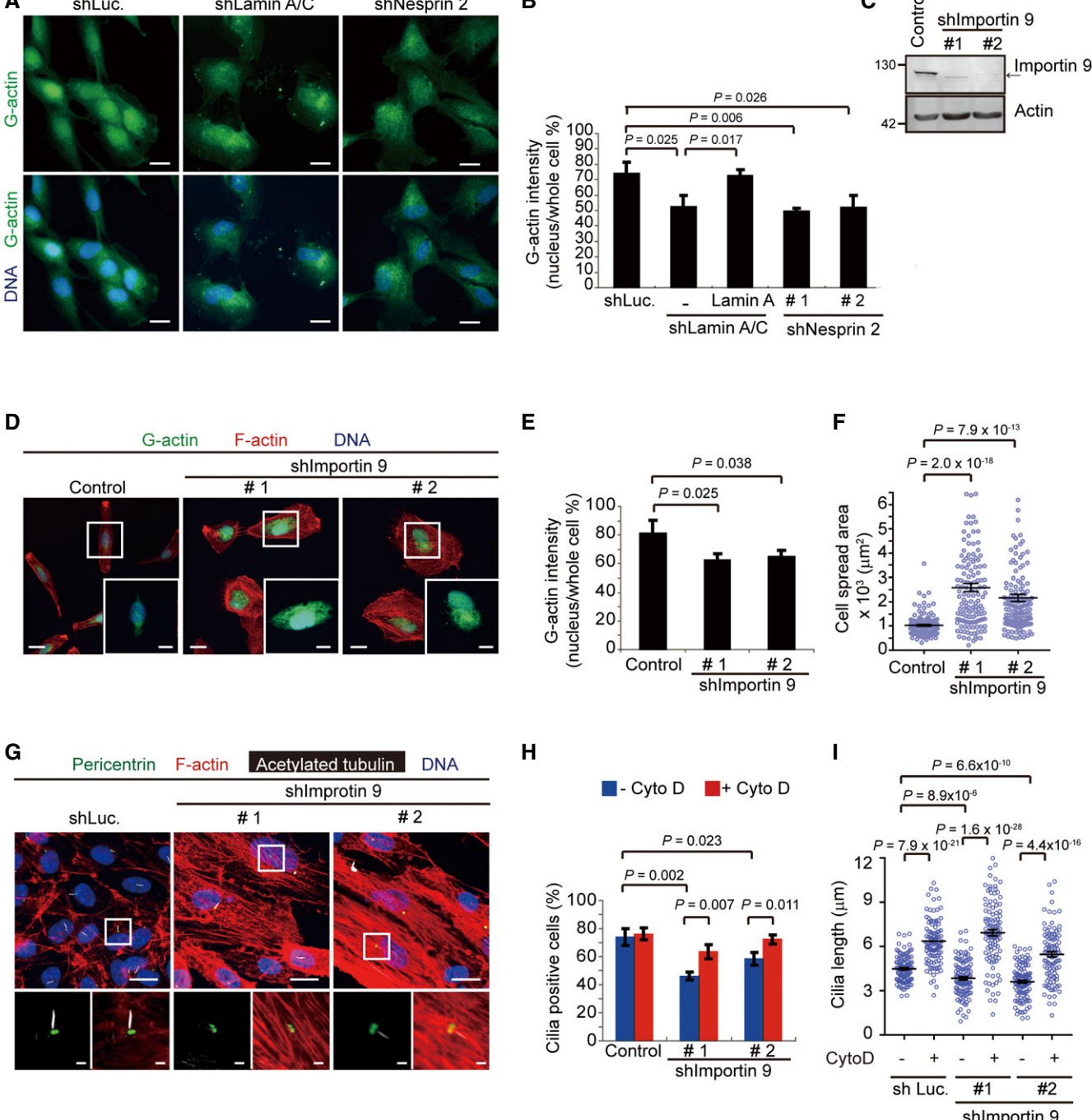

Figure 6.

◀

**Figure 6.  Deficiency in ciliogenesis by depletion of lamin A/C or nesprin 2 may be caused by disruption of the actin homeostasis.**

A, B  RPE cells expressing shRNAs specific to lamin A/C or nesprin 2 (shNesprin 2, clones #1, and #2) were stained for G-actin with DNase I (green). Representative images are shown (A). Scale bars, 20 μm. The proportion of nuclear G-actin fluorescence intensity to the whole cell was measured (B, $n \geq 67$).

C  An equal amount of whole-cell lysates from RPE cells (control) and those expressing shRNAs to importin 9 (shImportin 9, clones #1, and #2) was analyzed by immunoblotting with antibodies as indicated.

D–F  The cells expressing shRNAs to importin 9 were stained for G-actin (green) and F-actin (red). The representative images are shown (D). Scale bars, 20 or 10 μm (insets). The proportion of nuclear G-actin intensity to the whole cell (E, $n \geq 152$) and the cell spreading area (F, $n \geq 139$) were measured.

G–I  RPE cells and those expressing shRNAs to importin 9 were serum-starved for 32 h and then treated with (+) or without (−) Cyto D for another 16 h and then stained for acetylated tubulin (white), pericentrin (green), and F-actin (red). Representative images are shown (G). The left insets show the merged signals of acetylated tubulin and pericentrin. Scale bars, 20 or 2 μm (magnified images). The percentage of the cells with cilia (H, $n \geq 385$) and the length of cilia (I, $n \geq 111$) were measured.

Data information: In B, E and H, values (means ± SD) are from three independent experiments. In F and I, values (means ± SEM) are from three independent experiments. Statistical significance of differences is assessed with Student's *t*-test.

(Fig 8B and C, right). The expression levels of TTBK2 and CP110 were not affected by lamin A/C depletion (Appendix Fig S15). However, in non-ciliated cells, lamin A/C depletion caused decreases in TTBK2-recruitment and CP110-removal (Fig 8B and C, left). The defects in TTBK2-recruitment and CP110-removal were restored by disruption of F-actin by cytochalasin D (Fig 8B and C, left). These data together suggest that increased F-actin by lamin A/C depletion may somehow suppress the recruitment of TTBK2 to the mother centriole, which thereby fails to remove CP110 from the mother centriole.

## Discussion

In this study, we employed three systems including HGPS fibroblasts, *Lmna* null mice, and lamin A/C-depleted RPE cells to demonstrate an important role of lamin A/C in ciliogenesis. Our results indicate that the expression of progerin or suppression of lamin A/C expression impaired ciliogenesis, leading to decreased ciliation and cilia length. Likewise, the depletion of nesprin 2 had an adverse effect on ciliogenesis, similar to that caused by depletion of lamin A/C. Because lamin A/C and nesprin 2 are major components of the nuclear lamina, our data support that the nuclear lamina integrity is important for proper ciliogenesis. In our efforts to understand the underlying mechanism, we found that the disturbance of the actin homeostasis caused by the loss of the nuclear lamina integrity may be responsible for the impaired ciliogenesis. It is known that a large portion of the G-actin is stored in the nucleus (Gonsior *et al*, 1999; Huff *et al*, 2004; Grosse & Vartiainen, 2013). The disruption of nuclear lamina and the subsequent loss of the nuclear envelope integrity may disturb the G-actin equilibrium between the nucleus and cytoplasm, leading to accumulation of G-actin in the cytoplasm, where it increasingly polymerizes into F-actin. Excessive F-actin has been proposed to suppress ciliogenesis (Yan & Zhu, 2012). Consistent with this notion, we found that the disruption of F-actin by cytochalasin D restored the defects in ciliogenesis caused by depletion of lamin A/C or nesprin 2.

Recently, the centrosome is proposed as an actin-organizing center (Farina *et al*, 2015). The centrosome-surrounded branched F-actin is required for myosin Va to transport PCV to the mother centriole (Wu *et al*, 2018). In this study, we showed for the first time that Arp2 localized exclusively at the mother centriole in the control RPE cells. Depletion of lamin A/C or nesprin 2 caused Arp2 to localize at both mother and daughter centrioles, which led to

diffuse distribution of myosin Va around the mother and daughter centrioles and failure in ciliogenesis. These data suggest that precise trafficking of Arp2 and myosin Va-PCV to the mother centriole may be required for successful fusion of PCV into CV at the basal body (a model depicting in Fig EV5). Failure in these steps hampers ciliogenesis. It is thus possible that abundant Arp2/3-mediated polymerization of F-actin around the centrosome may prevent the precise trafficking of myosin Va-PCV to the basal body. Moreover, our results showed that, even prior to serum starvation, Arp2 already localized to both mother and daughter centrioles in lamin A/C-depleted cells. These data raise a possibility that loss of nuclear lamina integrity may somehow cause aberrant centrosomal duplication, leading to mislocalization of certain mother centriolar protein (s) to the daughter centriole, leading to Arp2/3 localization at both centrioles. However, the mechanism of how Arp2 localizes specifically to the mother centriole remains to be investigated.

Although TTBK2 is known to be essential for initiating cilium assembly (Goetz *et al*, 2012), the mechanism of how it localizes to the right place (at the basal body) in the right time (after CV docking) remains an enigma. Previous studies show that CEP164, a distal-appendage protein of the mother centriole, is required for recruiting TTBK2 to the basal body (Cajanek & Nigg, 2014). In addition, CEP164 is necessary for the docking of ciliary vesicles by interacting with the vesicular transporters, GTPase Rab8 and its guanine nucleotide exchange factor Rabin8 (Schmidt *et al*, 2012). In this study, we found that although the localization of CEP164 at the mother centriole was not affected by the depletion of lamin A/C or nesprin 2, TTBK2 was less localized to the mother centriole. These data indicate that CEP164 by itself is not sufficient to recruit TTBK2 to the basal body. Because TTBK2 localizes to the basal body after CV docking, it is possible that certain proteins on the CV may interact with TTBK2 and thereby recruit it to the basal body. For example, EHD1, a vesicle-shaping protein responsible for fusing PCVs into a single CV, was shown to be required for CP110-removal (Lu *et al*, 2015). It will be of interest to examine whether EHD1 interacts with TTBK2 and thereby removes CP110.

Previous studies showed that the mice harboring a *Lmna* L52R mutation exhibit impaired cilia in their middle ears and hearing loss (Odgren *et al*, 2010; Zhang *et al*, 2012). In this study, we showed that *Lmna* null mice, a model for EDMD (Sullivan *et al*, 1999), apparently form fewer and shorter cilia in their skeletal muscles than their control counterparts. In addition, *Lmna* null mice revealed different degrees of ciliary defects in the uterus, ovary, and kidney, which are organs commonly affected in ciliopathies

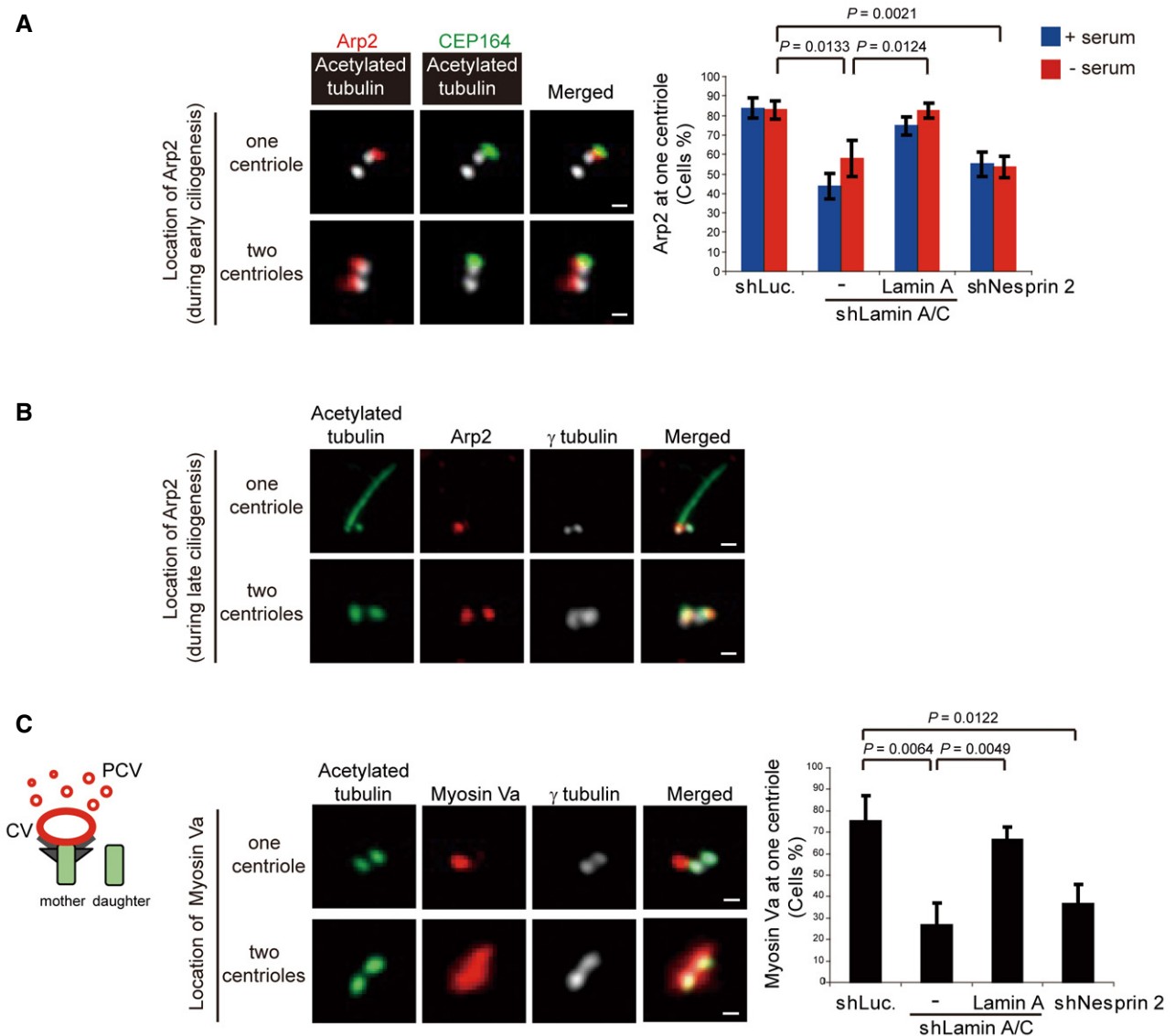

**Figure 7.  Depletion of lamin A/C or nesprin 2 causes a defect in docking of Arp2 and myosin Va-mediated preciliary vesicles to the basal body.**

A   RPE cells expressing shRNAs specific to luciferase (shLuc.), lamin A/C, or nesprin 2 were grown in the growth medium (+ serum) or serum-starved for 1 h (− serum) and then stained for Arp2 (red), CEP164 (a marker for the mother centriole, green), and acetylated tubulin (white). The representative images are shown. Note that Arp2 was found to localize exclusively at the mother centriole (upper panels) or at the both mother and daughter centrioles (lower panels). Scale bars, 0.5 μm. The percentage of cells with Arp2 localized exclusively at the mother centriole in the total counted cells was measured ($n$ = 131–240 cells in each group).

B   The cells were serum-starved for 48 h and then stained for acetylated tubulin (green), Arp2 (red), and γ-tubulin (white). The representative images are shown. Scale bars, 1 μm (upper panels) and 0.5 μm (lower panels). Note that the cell with Arp2 at the both mother and daughter centrioles fails to form cilium.

C   The cells were serum-starved for 48 h and then stained for myosin Va (red), acetylated tubulin (green), and γ-tubulin (white). The cartoon illustrates that myosin Va-associated vesicles (CV+PCV) specifically accumulate at the mother centriole in most control cells (as in the "one centriole" image). CV, ciliary vesicle. PCV, periciliary vesicle. The percentage of cells with myosin Va exclusively at the mother centriole in the total counted cells was measured ($n$ = 160–200 cells in each group).

Data information: In A and C, values (means ± SD) are from three independent experiments. Statistical significance of differences is assessed with Student's $t$-test.

(Fliegauf *et al*, 2007; Forsythe & Beales, 2013). However, the adverse effect of lamin A deficiency on ciliogenesis was more apparent in muscle and uterus than in kidney and ovary of *Lmna*$^{-/-}$ mice (Fig 2). In addition, the motile cilia lining the oviducts and bronchi appeared normally in *Lmna*$^{-/-}$ mice (Appendix Fig S5). These results strongly suggest that the significance of lamin A in ciliogenesis is likely to be tissue-dependent. This study raises an intriguing

possibility that laminopathies may be partially attributed to deficient primary cilia. In fact, patients of laminopathies and ciliopathies share some common phenotypes, such as diabetes, craniofacial abnormalities, hearing loss, short stature, and abnormal fat and bone tissues. However, the hallmark ciliopathy phenotypes, such as polycystic kidney or retinopathy, were not observed in *Lmna*$^{-/-}$ mice. It may be due to the early death of *Lmna*$^{-/-}$ mice. It is known

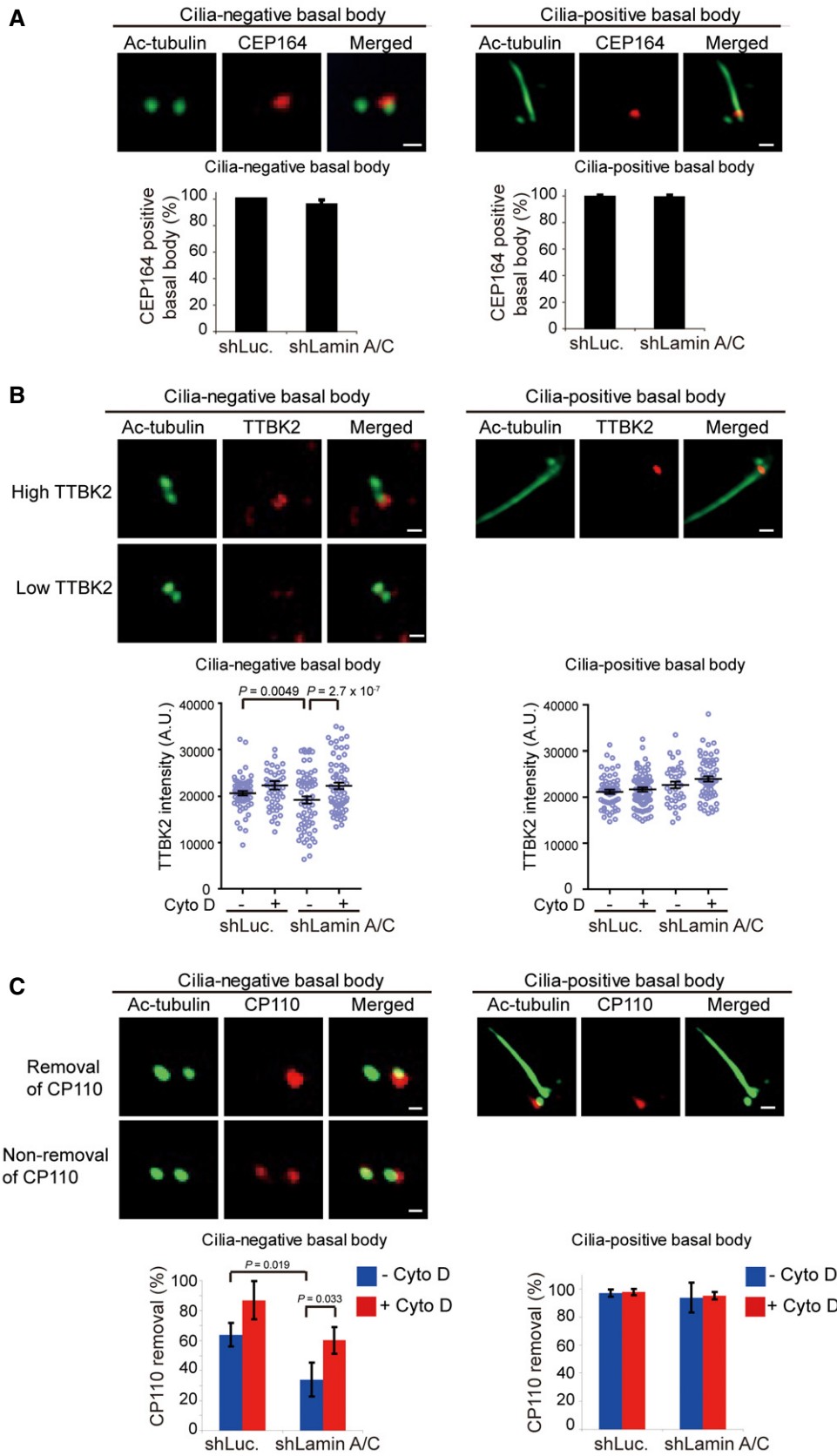

**Figure 8.**

**Figure 8.  Depletion of lamin A/C suppresses the recruitment of TTBK2 to the mother centriole.**

A  RPE cells expressing shRNAs specific to luciferase (shLuc) or lamin A/C (shLamin A/C) were serum-starved for 48 h and then stained for acetylated tubulin (Ac-tubulin, green) and CEP164 (red). The representative images show that the fluorescence intensity of CEP164 at the mother centriole of cilia-negative (left column) and cilia-positive (right column) basal bodies are similar. The graphs shown are the quantification of CEP164 at cilia-negative basal bodies and cilia-positive basal bodies.

B  The cells were serum-starved for 32 h and then treated with (+) or without (−) Cyto D for another 16 h. The cells were stained for acetylated tubulin (Ac-tubulin, green) and TTBK2 (red). The graphs shown are the quantifications of TTBK2 intensity at cilia-negative basal bodies and cilia-positive basal bodies.

C  The cells as in (B) were stained for acetylated tubulin (Ac-tubulin, green) and CP110 (red). The representative images show that CP110 is not removed from the mother centriole at some of cilia-negative basal bodies (left column), whereas it is removed from the mother centriole at cilia-positive basal bodies (right column). The graphs shown are the quantifications of CP110-removal at cilia-negative basal bodies and cilia-positive basal bodies.

Data information: Scale bars, 0.5 μm (A–C, panels from cilia-negative basal body); 1 μm (A–C, panels from cilia-positive basal body). In A and C, values (mean ± SD) are from three independent experiments. In B, values (means ± SEM) are from three independent experiments. $n \geq 60$ in each group. Statistical significance of differences is assessed with Student's t-test.

that polycystic kidney disease occurs at 30–40 years of age in humans. The medial life span of $Lmna^{-/-}$ mice is only 42 days, which is equivalent to 6 years of age in humans. Therefore, it is possible that $Lmna^{-/-}$ mice die before the appearance of the ciliopathy phenotypes.

In this study, we propose that the nuclear lamina integrity is important for proper ciliogenesis through guarding the actin homeostasis. In addition to lamin, nesprin 2 and other LINC proteins, Sun1 and emerin, have been implicated to involve in the pathogenesis of progeria or EDMD of the laminopathies (Kandert et al, 2007; Zhang et al, 2007; Chen et al, 2012; Ho et al, 2013). Because emerin functions along with lamin A/C in modulating actin dynamics (Ho et al, 2013) and Sun1 is required for stabilizing the nuclear pore complex (Liu et al, 2007a), both of which may also contribute to maintain the nucleus-cytoplasm shuttling of G-actin. Therefore, it will be of interest to determine whether Sun1 and emerin also participate in ciliogenesis. Moreover, increasing evidence indicates that many molecules are localized to both the cilium and nucleus (Liu et al, 2007b; Ishikawa et al, 2012; Del Viso et al, 2016). For example, lamin B1 was identified as a candidate ciliary protein from proteomic analyses of primary cilia (Liu et al, 2007b; Ishikawa et al, 2012). In addition, some of the nuclear pore complex proteins are found at cilia bases and function for cilia (Del Viso et al, 2016). A comprehensive survey of whether lamins and lamin-associated proteins are localized in the cilium and have a direct role in regulating ciliogenesis awaits for future research efforts.

## Materials and Methods

### Antibodies and reagents

The anti-progerin (ab66587), anti-lamin A (ab8980), anti-aquaporin 1 (ab65837), anti-Arl3b (ab136648), and anti-pericentrin (ab4448) antibodies were purchased from Abcam. The anti-lamin A/C antibody (for immunoblotting, customized) was generated by GeneTex (Taiwan). The anti-emerin (sc-15378), anti-Arp2 (sc-15389), anti-WASP (sc-8353), and anti-CEP164 (used in Fig 7A, sc-515403) antibodies were purchased from Santa Cruz Biotechnology. The anti-acetylated tubulin (T6793), anti-γ-tubulin (T5326), anti-TTBK2 (HPA018113), anti-actin (A5441), and anti-α–tubulin (T6199) antibodies were purchased from Sigma-Aldrich. The anti-nesprin 2 (PA5-51933) and anti-Arp3 (PA5-80307) antibodies were purchased from Thermo Fisher Scientific. The anti-CEP164 antibody (used in Fig 8A, 4533.00.02) was purchased from SDIX. The anti-myosin Va antibody (NBP1-92156) was purchased from Novus. The anti-CP110

antibody (12780-1-AP) was purchased from ProteinTech. The anti-keratin 8 antibody (AB 531826) was purchased from DSHB. The anti-centrin antibody (04-1624) was purchased from Merck Millipore. Phalloidin (conjugated with Alexa Fluor-488 or -546), DNase I (conjugated with Alexa Fluor-488), and fetal bovine serum (FBS) were purchased from Invitrogen. Cytochalasin D was purchased from Sigma-Aldrich. SAG was purchased from Enzo.

### Mice, immunohistochemistry and measurement of cilia length in tissues

The $Lmna^{-/-}$ mice were generated and described previously (Sullivan et al, 1999). The 4-week-old $Lmna^{-/-}$ mice and their control littermates ($Lmna^{+/+}$ and $Lmna^{+/-}$) were obtained from the Jackson Laboratory. The animal care and experimental procedures using mice were approved by the Institutional Animal Care and Use Committee (IACUC) at National Health Research Institutes. Tissues were collected and processed for histological analysis as described previously (Chen et al, 2016). To image the tilted cilia in the tissue, the Z-slices of a 0.15-μm interval were taken under a 100× objective. To measure the cilia length $c$, the length of cilia on the maximum intensity projection (MIP) was used as $a$, and the numbers and thickness of Z-slices provided as $b$. The cilia length $c$ was then calculated according to the Pythagorean Theorem by the formula: $c^2 = a^2 + b^2$.

### Cell culture

RPE cells were cultured in Dulbecco's modified Eagle's medium supplemented with 10% FBS. HGPS human skin fibroblasts (AG01972, AG08466, AG11513, AG06297, AG06917) and normal control fibroblasts (AG03512, AG03257) were described previously (Chen et al, 2012) and obtained from the National Institute on Aging, Aged Cell Repository at Coriell Institute. MEFs were prepared from E15.5 embryos. Cells were dissociated by trypsin at 37°C for 30 min, passed through 40-μm cell strainer (Falcon). Human skin fibroblasts and MEFs were maintained in Eagle's minimum essential medium (Thermo Fisher Scientific) supplemented with 15% fetal bovine serum (HyClone, Characterized), 2 mM ʟ-glutamine, and antibiotics.

### Lentivirus-mediated gene knockdown and expression

The lentiviral expression system was obtained from the National RNAi Core Facility (Academia Sinica, Taiwan). The target sequence for lamin A was 5′-GCCGTGCTTCCTCTCACTCAT-3′ (#1). The target

sequences for nesprin 2 were 5′-CCTCAGTTATATCCGACCTAT-3′ (#1) and 5′-GCCACCTATGAGTCTGTCAAT-3′ (#2). The target sequences for importin 9 were 5′-GAGGATTACTACGAGGATGAT-3′ (#1) and 5′-CCTGACAACAGTAGTACGAAA-3′ (#2). For the expression of lamin A/C, the full-length lamin A cDNA was amplified by polymerase chain reaction and cloned into the pLAS3w.Pneo vector. Lentiviral production and infection were performed as described previously (Chan *et al*, 2014). RPE cells were infected with lentiviruses encoding shRNAs or cDNA for 24 h, and then selected with puromycin (2 μg/ml) or G418 (600 μg/ml), respectively. The RPE cells expressing shRNAs were analyzed between 5 and 10 days after viral infection.

### Generation of lamin A/C-knockout RPE cells

RNA-guided DNA endonuclease was performed to edit genes through co-expression of the Cas9 protein (Addgene plasmid 41815) with gRNAs (http://www.addgene.org/crispr/church/). The targeting sequence for lamin A/C was 5′-GGAGCTCAATGATCGCTTGG-3′. The target sequences were cloned into the gRNA cloning vector (Addgene plasmid 41824) via the Gibson assembly method (New England Biolabs). Lamin A/C-knockout cells were obtained through clonal propagation from a single cell. For genotyping, the following PCR primers were used: 5′-CGCACCTACACCAGCCAA-3′ and 5′-CGAACTCACCGCGCTTTC-3′. PCR products were cloned and sequenced.

### Immunofluorescence staining, microscopy, and image analysis

For the co-staining of acetylated tubulin with CP110, TTBK2, CEP164, Arp2, Arp3, WASP, myosin Va, or γ-tubulin, cells were fixed with 100% methanol at −20°C for 10 min. Otherwise, cells were fixed with phosphate-buffered saline containing 4% paraformaldehyde at room temperature for 15 min. The cells were permeabilized with 0.1% Triton X-100 for 60 min or 0.5% Triton X-100 for 20 min after methanol- or paraformaldehyde-fixation, respectively. The fixed cells were stained with primary antibodies for 1 h then incubated with Alexa Fluor 488- or 546-conjugated secondary antibodies, phalloidin, or DNase I for 1 h. Coverslips were mounted on the slides with mounting medium (Anti-Fade Dapi-Fluoromount-G, Southern Biotech). The images were acquired using an upright fluorescence microscope (Axio imager, M2 Apotome2 system, Carl Zeiss), which was equipped with 63× and 100× oil-immersion objective lens (1.4 NA, Plan Apochromat) and a camera (ORCA-Flash4.0V2; Hamamatsu). The acquired images were processed and analyzed by the ZEN2 software (Carl Zeiss). The cilia length was manually measured according to the signals ranging from the r-tubulin-labeled basal body to the acetylated tubulin-labeled axonemal tip, or according to the Arl13b signal. The images (2,048 × 2,048 pixels) were taken by a 63× oil-immersion objective (1.4 NA, Plan Apochromat) and analyzed by the ZEN2 software (Carl Zeiss). The representative images were cropped by Photoshop CS5 (Adobe), assembled by Illustrator CS5 (Adobe).

### Immunoblotting

To prepare whole-cell lysates, cells were lysed on ice in RIPA buffer (0.1% SDS, 1% sodium deoxycholate, 1% NP-40, 150 mM NaCl, 50 mM Tris–HCl, 1 mM EDTA, pH 7.4). Immunoblotting was performed as previously described (Hsu *et al*, 2018).

### Cellular fractionation

Cellular fractionation was performed using the NE-PER™ nuclear cytoplasmic protein separation kit (Thermo Fisher Scientific) according to the manufacturer's instructions. An equal proportion of the lysates from the nuclear and cytoplasmic fractions were analyzed by immunoblotting with specific antibodies. Lamin A/C and α-tubulin were used as markers for the nuclear and cytoplasmic fractions, respectively. The relative ratio of actin in the nuclear and cytoplasmic fractions was quantified using the Image Studio Lite software.

### Flow cytometry

Cells were dissociated by trypsin-EDTA, washed with PBS, and fixed with absolute ethanol (−20°C). After incubating with RNaseA (20 μg/ml), cells were stained with propidium iodide (10 μg/ml) and read on a Becton Dickinson FACSCalibur. The data were analyzed with the FlowJo-V10 software.

### Quantitative RT–PCR

Total cellular RNA was isolated using Quick-RNA™ MiniPrep Kit (Zymo Research) and then digested with DNase I to remove genomic DNA. The cDNA was generated using RevertAid First Strand cDNA Synthesis Kit (Thermo). The quantitative PCR was performed on an Applied Biosystems instrument using Maxima SYBR Green/ROX qPCR Master Mix (2×) (Thermo). The primers for Gli1 and RPL19 (as the internal control) were as follows: Gli1 forward primer 5′-GAGCGGAAGGAATTCGTGTG-3′, Gli1 reverse primer 5′-TGGGATCTGTGTAGCGCTTG-3′, RPL19 (60S) forward primer 5′-GAAATCGCCAATGCCAACTC-3′, and RPL19 (60S) reverse primer 5′-TCCTTGGTCTTAGACCTGCG-3′.

### Statistics

The two-tailed Student's *t*-test was used to determine whether the differences between experimental values were considered significant. $P < 0.05$ was considered statistically significant. The number of cells counted for each experiment was written in the respective figure legend.

# Data availability

We have not generated data that require deposition in a public database.

**Expanded View** for this article is available online.

### Acknowledgements

This work was supported by the Ministry of Science and Technology, Taiwan (grant number 106-2320-B-005-011-MY3, 107-2923-B-005-002-MY3, and 108-2320-B-010-015-MY3) and the Cancer Progression Research Center, National Yang-Ming University from the Featured Areas Research Center Program

                                          

within the framework of the Higher Education Sprout Project by the Ministry of Education (MOE) in Taiwan.

## Author contributions

HCC designed the research project. JRF performed the experiments and analyzed the data; LRY performed tissue processing and immunohistochemistry. WJW contributed to epifluorescence microscopy and discussion; WSH contributed to cilia measurement. CTC contributed to cellular fractionation and figure preparation; YHC maintained and provided *Lmna* null mice, MEFs and HGPS fibroblasts; JRF and HCC wrote the manuscript.

## Conflict of interest

The authors declare that they have no conflict of interest.

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
