## [Review Process File · EMBO Reports]

Lamin A-mediated nuclear lamina integrity is required for proper ciliogenesis

Jia-Rong Fan, Li-Ru You, Won-Jing Wang, Wei-Syun Huang, Ching-Tung Chu, Ya-Hui Chi and Hong-Chen Chen

DOI: [10.15252/embr.201949680](https://doi.org/10.15252/embr.201949680)

Corresponding author(s): Hong-Chen Chen (hcchen1029@ym.edu.tw)

Review Timeline:

Submission Date:	14th Nov 19
Editorial Decision:	22nd Jan 20
Revision Received:	21st Apr 20
Editorial Decision:	5th Jun 20
Revision Received:	12th Jul 20
Accepted:	23rd Jul 20

Transaction Report:

Dear Prof. Chen

Thank you for the submission of your research manuscript to our journal. I apologize for the delay in handling your manuscript, but we have only recently received the full set of referee reports, which is copied below.

As you will see, the referees acknowledge that the findings are potentially interesting but they also indicate that a major revision will be required to substantiate the findings. The experimental details will need to be better documented and described, the data on cilia phenotypes need better documentation and higher quality pictures and it will be important to extend the description to more hallmarks of ciliopathies. Moreover, the link between laminopathies and ciliopathies needs to be substantiated and the analysis extended to tissues and cells of the *Lmna* null mouse and to patient fibroblasts.

From these comments it is clear that a major revision will be required before. However, given the constructive comments, we would like to invite you to revise your manuscript with the understanding that the referee concerns (as detailed above and in their reports) must be fully addressed and their suggestions taken on board. Please address all referee concerns in a complete point-by-point response. Acceptance of the manuscript will depend on a positive outcome of a second round of review. It is EMBO reports policy to allow a single round of revision only and acceptance or rejection of the manuscript will therefore depend on the completeness of your responses included in the next, final version of the manuscript.

Revised manuscripts should be submitted within three months of a request for revision; they will otherwise be treated as new submissions. Please contact us if a 3-months time frame is not sufficient for the revisions so that we can discuss the revisions further.

- 1) A data availability section providing access to data deposited in public databases is missing (if relevant).
- 2) Your manuscript contains error bars based on $n=2$. Please use scatter blots showing the individual datapoints in these cases. The use of statistical tests needs to be justified.

2) individual production quality figure files as .eps, .tif, .jpg (one file per figure).

Please download our Figure Preparation Guidelines (figure preparation pdf) from our Author Guidelines pages

<https://www.embopress.org/page/journal/14693178/authorguide> for more info on how to prepare your figures.

4) a complete author checklist, which you can download from our author guidelines (). Please insert information in the checklist that is also reflected in the manuscript. The completed author checklist will also be part of the RPF.

5) Please note that all corresponding authors are required to supply an ORCID ID for their name upon submission of a revised manuscript (). Please find instructions on how to link your ORCID ID to your account in our manuscript tracking system in our Author guidelines ()

6) We replaced Supplementary Information with Expanded View (EV) Figures and Tables that are collapsible/expandable online. A maximum of 5 EV Figures can be typeset. EV Figures should be cited as 'Figure EV1, Figure EV2' etc... in the text and their respective legends should be included in the main text after the legends of regular figures.

7) We would also encourage you to include the source data for figure panels that show essential data. Numerical data should be provided as individual .xls or .csv files (including a tab describing the data). For blots or microscopy, uncropped images should be submitted (using a zip archive if multiple images need to be supplied for one panel). Additional information on source data and instruction on how to label the files are available .

8) Our journal encourages inclusion of *data citations in the reference list* to directly cite datasets that were re-used and obtained from public databases. Data citations in the article text are distinct from normal bibliographical citations and should directly link to the database records from which the data can be accessed. In the main text, data citations are formatted as follows: "Data ref: Smith et al, 2001" or "Data ref: NCBI Sequence Read Archive PRJNA342805, 2017". In the Reference list, data citations must be labeled with "[DATASET]". A data reference must provide the database name, accession number/identifiers and a resolvable link to the landing page from which the data can be accessed at the end of the reference. Further instructions are available at .

9) Regarding data quantification:

- Please ensure to specify the name of the statistical test used to generate error bars and P values, the number (n) of independent experiments underlying each data point (not replicate measures of one sample), and the test used to calculate p-values in each figure legend. Discussion of statistical methodology can be reported in the materials and methods section, but figure legends should

contain a basic description of n, P and the test applied.

IMPORTANT: Please note that error bars and statistical comparisons may only be applied to data obtained from at least three independent biological replicates. If the data rely on a smaller number of replicates, scatter blots showing individual data points are required and the statistical test used needs to be justified.

- Graphs must include a description of the bars and the error bars (s.d., s.e.m.).
- Please also include scale bars in all microscopy images.

10) As part of the EMBO publication's Transparent Editorial Process, EMBO reports publishes online a Review Process File to accompany accepted manuscripts. This File will be published in conjunction with your paper and will include the referee reports, your point-by-point response and all pertinent correspondence relating to the manuscript.

I look forward to seeing a revised version of your manuscript when it is ready. Please let me know if you have questions or comments regarding the revision.

Yours sincerely

Martina Rembold, PhD
Editor
EMBO reports

Referee #1:

The authors present interesting data, indicating that lamin a/c depletion affects ciliogenesis. This is in line with previous findings for a functionally related protein, nesprin 2 (or Syne-2), although not all experiments presented are described in sufficient details. The authors also propose a model that implies a role of the nuclear lamina integrity in ciliogenesis, and suggest that malfunction of primary cilia may be involved in the pathogenesis of laminopathies. Both of these conclusions, although interesting, are not yet sufficiently well supported, and the authors need to include additional experimental evidence to substantiate them, or tone down their statements and discuss the alternative options. The rationale for this criticism is indicated point-by-point below.

1.) Overall, the manuscript is well-written, but the Materials and Methods section is in general too brief and lacks many important experimental details that are required to evaluate the relevance of the findings. These are indicated with the more specific comments below. Also, the abstract and introduction contain several erroneous or incomplete statements. The most important ones:
Abstract: "[The] Primary cilium...: "the" omitted ... "detects changes in the extracellular environment":

this is not its ubiquitous role; it is receiving specific signals from the extracellular environment, sometimes detecting changes, mostly not. It not just "regulates various processes" this is too broad and can be stated for almost any cellular machinery. What are "Arp2 and myosin Va-mediated ciliary vesicles? They may not be known by the reader. Why is failure in the recruitment of tau tubulin kinase 2 relevant? Without these indications, the abstract is incomprehensive.

Introduction: First word: "[The] Primary cilium...: "the" omitted ...; Same sentence: "harbors signalling receptors" a specific set, cell-type specific, required for key developmental processes such as Hedgehog signalling.

The fourth sentence (P. 3, lines 5-7) is incorrect: they don't all rely on primary cilia for all of these processes; include "respectively". The phenotypes given for ciliopathies (page 3, lines 8-10) seem random and are incomplete; these are not the main hallmark phenotypes.

P. 3, line 4 from bottom: what are "other types of the cytoskeleton"?

Reference 39: doesn't mention impaired cilia.

2.) The cilia in many of the figures (e.g. Fig. 1a, c, d; Fig. 4d; Fig. 5c; Fig. 6g) are very hard to evaluate: the images lack detail and as mostly overlays are shown, the signals from different individual channels are often very hard to distinguish. For these cases, at least provide figures with separated channels and multiple insets of representative cilia as supplements, so the reader can evaluate the data on which the measurements and graphs are based.

Supplementary Fig. S1 is of insufficient quality to evaluate if the green signal actually detects motile cilia, as they cannot be distinguished. The conclusion that motile cilia are not affected by lamin A depletion is therefore not warranted without improved imaging.

3.) Details of the ciliary length measurements are lacking: were these done manually, or automated. What parameters were used? What was the distribution? Just showing bar graphs is not sufficient to represent the cilium length distribution. Were biological or technical replicates used? What was the confluency of the cultures, which is a critical parameter that affects cilium frequency?

4.) Was the cell cycle and/or cell viability affected by the different shRNA treatments? These parameters are commonly evaluated using FACS analysis.

5.) Fig. 1A: why do only some patient-derived fibroblasts have high progerin levels? This remains undiscussed but is critical to the results.

6.) Why do high levels of progerin show the same effect as lamin A/C depletion? What is the range? This should be discussed.

7.) The authors do not show the increased formation of actin filaments, nor any of the other cellular phenotypes (except from decreased cilium length) in the different tissues of the Lmna null mice. Do these mice not show these phenotypes? And why is only cilium length indicated? Is cilium frequency not affected? This should be indicated. If the indicated cellular phenotypes are not visible in the knockout mice, this is essential to the references to laminopathies as well as ciliopathies, and should be critically assessed and discussed.

8.) The authors show that deficiency of lamin A in the mice cause cilium length defects, and relate that to a role of cilia in laminopathies. However, they don't show a cystic kidney phenotype in these mice, nor do they mention other hallmark ciliopathy phenotypes such as retinopathy or L/R randomization. Were these not observed? This should be evaluated and discussed.

9.) As the authors have access to Lmna null mice, why were these not deployed to evaluate the

embryonic fibroblasts for the characteristics where they now use shRNA depletion instead? These are excellent cells for imaging and widely used to evaluate ciliopathy-associated mechanisms. Validating at least a subset of the cellular phenotypes in these cells would significantly strengthen the story, and would also allow evaluation of signalling pathways such as Hedgehog signalling and mTOR signalling, that are commonly affected in ciliopathies. Even the effect of potential drugs such as rapamycin, which they mention in their discussion, could then be easily evaluated.

10.) The only rationale for a relationship between nuclear lamina integrity and ciliogenesis is the G-actin experiment, although this relationship could be very well indirect: increased F-actin could trigger many defects in the cell, several of which could contribute to ciliogenesis defects, such as mislocalization of many actin binding proteins that are important for cilium function and ciliogenesis. Also in light of this, it is important to evaluate if the cell cycle is affected, and if the actin stress fibers/F-actin activation and cellular remodelling are also found in the patient cells and in the *Lmna* null mice .

11.) Arp2 localization is used as a measure for actin dynamics activity, but it seems to remain strictly localized to the centrioles. It is therefore important to also evaluate other components of this machinery (the Arp2/3 complex contains seven subunits of which Arp2 is just one, and actin dynamics involves many potential contributors) to validate these findings.

12.) The downregulation of Nesprin 2 is a surprising finding and was suggested to be related to lamin A/C function mechanistically. Therefore, it is important to show (validate) and discuss if this downregulation is consistent in the patient cells and *Lmna* null mice. Also, Dawe et al. (Ref. 38) showed that Nesprin 2 associates with Meckelin, a ciliopathy-associated protein, and that depletion of Meckelin leads to reduction of Nesprin 2, similar to lamin A/C. Interestingly, this correlated with an upregulation of activated RhoA, suggesting regulation via the ROCK-RhoA signalling pathway. In light of these earlier studies, it would be very informative to evaluate a potential activation of RhoA also in the lamin A depleted cells, and/or a potential co-depletion of meckelin.

In conclusion, the authors present potentially interesting findings that disruption of lamin A/C causes ciliogenesis and cilium length defects via F-actin activation, but many gaps still exist in their hypotheses. Also, the representation of several of their results is currently of insufficient quality. The authors should be able to address the indicated issues experimentally, as they do have all the material and reagents at hand. It would be most important to validate the results of actin filament formation and increase of cell spreading/cell remodelling, as well as the Arp2, TTBK2, CP110 and Myosin Va stainings , in the *Lmna* null mice or in embryonic fibroblasts derived thereof, and in the HSPS patient cells. This would significantly strengthen their hypothesis that the described defects resulting from lamin A/C dysfunction result in ciliogenesis defects that potentially play a role in laminopathies.

Referee #2:

This is an interesting manuscript (MS) that shows cells with depleted levels of lamin A/C or expression of an aberrant form of lamin A (progerin) have alterations in cilia incidence and length. Data linking the cilia abnormalities in lamin depleted cells to alteration in the actin cytoskeleton are presented. A potential mechanism where increased F-actin in response to lamin depletion leads to impaired recruitment of TTBK2 to the mother centriole, resulting in reduced disassociation of CP110 that represses ciliogenesis, is also described. While the observation that cilia incidence and

length is reduced in lamin depleted cells is relatively convincing I believe aspects of the MS could be strengthened through additional controls and experiments.

The following points should be considered:

1. Cilia quantification is most convincing in studies that use dual labeling with two cilia markers for conclusive identification of cilia. This could be through use of an additional axonemal marker e.g. an antibody against Arl13b or a basal body antibody e.g. against pericentrin or gamma tubulin (in addition to detection of acetylated tubulin). If dual labelling is not possible quantification in alteration of cilia incidence length could still be undertaken with a different antibody to further confirm results. I do not feel this is necessary for all experiments where cilia are quantified, but would be a good additional control for initial experiment in the study (such as those in Figure 1-3).
2. The authors conclude the reduction in cilia incidence and length is a consequence of impaired ciliogenesis. They have induced ciliogenesis through serum starvation for 32 hours. This is a relatively long time that would be predicted to put all cells in cultures into stationary phase. In addition to ciliogenesis, changes in cilia dynamics may be a consequence of altered cilia resorption. The authors have not demonstrated conclusively that the changes they observe are solely a consequence of altered cilia formation. While this may be challenging to fully test, the conclusion would be strengthened by inclusion of data quantifying cilia formation over time after serum withdrawal (e.g. 2, 4, 8, 12, 16, 24, 32 hours).
3. In the experiments quantifying cilia incidence and length in HGPS fibroblast, shown in Figure 1, cells are classified as having high or low progerin expression. There is a big variability of progerin levels in the high progerin cells. Is the relationship between progerin levels and cilia length a linear inverse correlation across this range of progerin levels? This should be tested for the HGPS cell lines i.e. progerin levels vs cilia length should be plotted.
4. For the quantification of actin filaments within $60 \mu\text{m}^2$ of the basal body, how was the basal body position identified. No labelling for basal body is shown in the immunofluorescent labelling presented. Position of the basal body should be based on this labelling. It would also be useful to show on the representative images where the $60 \mu\text{m}^2$ area, in which actin filaments were quantified, is. Presumably this was a circle of $\sim 4.37 \mu\text{m}^2$ around the basal body?
5. The authors conclude disruption of F actin by cytochalasin D restores cilia assemble and length in high progerin cells. They do not present any images or quantification showing that the cytochalasin D treatment used reduced actin filaments around the basal body. This should be included.
6. In the Lmna null mouse tissues was there any change in primary cilia incidence compared to controls. The cilia length data is clear, but no comment is made on incidence.
7. The representative image presented for rescue of cilia in cells 're-expressing' lamin A does not support the quantitative data very well. Only 2 adjacent cells, of approximately 13, appear to have 'normal' cilia.
8. Figure 4. Parts d & e; similar to point 4 above, this data quantifies stress fibers in proximity to the basal bodies, yet basal bodies are not directly labelled. Antibody combinations that make this possible exist and it should be done.
9. For the nespín knockdown experiment in Figure 5, again stress fibers around the basal body are

quantified in cells where the basal body has not been immunolabelled.

10. The data on redistribution of G actin from the nucleus to the cytoplasm in lamin deficient cells is particularly interesting as it provides mechanistic insight into the observations of the MS. Is a redistribution of actin from the nucleus also observed in the HGPS fibroblasts and does this correlate with progerin levels? This should be tested. Can the re localization of actin from the nucleus to the cytoplasm be quantified by any other means to support the fluorescence intensity data? Subcellular fractionation followed by immunoblot or FRAP experiments would be options.

11. For figure 7 it would be more convincing if further examples of Arp2 localization were shown. Can low power images with multiple centrosomes/cilia be shown with zoomed panels for the centrioles. This is an important finding, but showing a single representative image and the quantification is not ideal. If there are space constraints in the MS this could be included in the supplemental.

12. In the results (page 8) the MS states that '... lamin A/C depletion caused ~30% decrease in the number of basal bodies with TTBK2-recruitment and CP-110 removal'. Is dual labelling possible to show that the basal bodies with increased TTBK2 are more likely to lose CP110? These results are presented and discussed in term of recruitment of TTBK2 and CP110 to cilia, immunoblots to show total cellular levels of these proteins are not changed by depletion of lamin should be included.

13. In the discussion the authors suggest that defective cilia function may be critical to the etiology of laminopathies. This argument would be strengthened if primary cilia function, rather than just structure, was analyzed in the lamin depleted cells. Shh signaling and other reporter assays are available to test this.

Minor comments

1. Labelling of lanes for the immunoblot presented in Figure 3a could be improved. Shorten control to 'Con' and or angle at 45o. Or even better show if each treatment/condition is +/- for each lane of the blot.

2. There are a number of badly constructed sentences and grammatical errors in the MS that need to be addressed.

3. The introduction and discussion could be more concise including a less speculative discussion.

Referee #1:

1. (1) Abstract: "[The] Primary cilium...: "the" omitted ... "detects changes in the extracellular environment": this is not its ubiquitous role; it is receiving specific signals from the extracellular environment, sometimes detecting changes, mostly not. It not just "regulates various processes" this is too broad and can be stated for almost any cellular machinery. (2) What are "Arp2 and myosin Va-mediated ciliary vesicles? They may not be known by the reader. Why is failure in the recruitment of tau tubulin kinase 2 relevant? Without these indications, the abstract is incomprehensive. (3) Introduction: First word: "[The] Primary cilium...: "the" omitted ...; Same sentence: "harbors signalling receptors" a specific set, cell-type specific, required for key developmental processes such as Hedgehog signalling. The fourth sentence (P. 3, lines 5-7) is incorrect: they don't all rely on primary cilia for all of these processes; include "respectively". The phenotypes given for ciliopathies (page 3, lines 8-10) seem random and are incomplete; these are not the main hallmark phenotypes. P. 3, line 4 from bottom: what are "other types of the cytoskeleton"? (4) Reference 39: doesn't mention impaired cilia.

Response:

- (1) The abstract has been revised according to the reviewer's comments.
- (2) The previous report (Wu et al., 2018) demonstrated that myosin Va mediates the transportation of preciliary vesicles (PCV) to the basal body via Arp2/3-associated actin network. Additionally, the formation of ciliary vesicles (CV) at the basal body is a prerequisite for TTBK2-recruitment to the basal body (Cajanek and Nigg, 2014; Schmidt et al., 2012). Therefore, we reason that the defect in the docking of myosin Va-mediated ciliary vesicles to the basal body will hamper the recruitment of TTBK2 to the basal body in lamin A-depleted cells.
- (3) The introduction has been revised according to the reviewer's comments.
- (4) The reference was corrected.

2. The cilia in many of the figures (e.g. Fig. 1a, c, d; Fig. 4d; Fig. 5c; Fig. 6g) are very hard to evaluate: the images lack detail and as mostly overlays are shown, the signals from different individual channels are often very hard to distinguish. For these cases, at least provide figures with separated channels and multiple insets of representative cilia as supplements, so the reader can evaluate the data on which the measurements and graphs are based. Supplementary

Fig. S1 is of insufficient quality to evaluate if the green signal actually detects motile cilia, as they cannot be distinguished. The conclusion that motile cilia are not affected by lamin A depletion is therefore not warranted without improved imaging.

Response:

- (1) For Fig 1A, the images with separated channels are shown in Appendix Fig S1.
- (2) For Fig 1C and D, the images with separated channels are shown in Appendix Fig S2
- (3) In new Fig 4D, 5C and 6G, the representative images of cilia are enlarged in the insets with merged signals of acetylated tubulin and pericentrin.
- (4) The old Supplementary Fig. S1 is replaced by Appendix Fig S5.

3. (1) Details of the ciliary length measurements are lacking: were these done manually, or automated. What parameters were used? (2) What was the distribution? Just showing bar graphs is not sufficient to represent the cilium length distribution. (3) Were biological or technical replicates used? (4) What was the confluency of the cultures, which is a critical parameter that affects cilium frequency?

Response:

- (1) To image the tilted cilia in the tissue, the Z-slices of a 0.15 μm interval were taken under a 100x objective. To measure the cilia length c , the length of cilia on the MIP (Maximum intensity projection) was used as a , and the numbers and thickness of Z-slices provided as b . The cilia length c was then calculated according to the Pythagorean Theorem by the formula: $c^2 = a^2 + b^2$. The cilia length was manually measured from the γ -tubulin-labeled basal body to the acetylated tubulin-labeled axonemal tip. In new Appendix Fig S3, S4, and S8, the cilia length was measured based on the signal of Arl13b, an axonemal marker. The images (2048 x 2048 pixels) were taken by a 63x oil-immersion objective (1.4 NA, Plan Achromat) and analyzed by the ZEN2 software (Carl Zeiss).
- (2) In new Fig 1F, 2E, 3D, 4G, 5F and 6I and Appendix Fig S3D, S4B, S6I and S8B, the original data of cilia length are plotted to show their distribution.
- (3) All quantitative data were obtained from at least three independent experiments and the numbers of samples enrolled in the experiments were indicated in figure legends.
- (4) To measure cilia frequency and length, the confluency of the cultures is always 90~95%.

4. Was the cell cycle and/or cell viability affected by the different shRNA treatments? These parameters are commonly evaluated using FACS analysis.

Response:

As suggested by the reviewer, we carried out FACS analysis and found that the cell cycle of RPE cells was not affected by the shRNAs used in this study (as shown below).

Figures for referees not shown.]

5. Fig 1A: why do only some patient-derived fibroblasts have high progerin levels? This remains undiscussed but is critical to the results.

Response:

The expression level of progerin is age-dependent in human/mice and cumulative with cell passages in HGPS patient-derived fibroblasts (Goldman et al., 2004; Chen et al., 2014). It is also well documented that only a portion of patient-derived fibroblasts have high-level progerin (Chen et al., 2012), which may due to the intrinsic nature of the primary cells with mixed population and various division cycles.

6. (1) Why do high levels of progerin show the same effect as lamin A/C depletion? (2) What is the range?

Response:

(1) Progerin is a persistently farnesylated form of lamin A, which has been shown to behave like a dominant negative mutant of lamin A (Eriksson et al., 2003; Capell et al., 2005). In other words, progerin will disturb the normal function of lamin A. This can partially explain why high levels of progerin show the same effect as lamin A/C depletion.

(2) The results in Fig EV1 show an inverse correlation between cilia length and progerin level in HGPS fibroblasts with progerin intensity above 3.0×10^6 A.U. (patient AG01972), 3.9×10^6 A.U. (patient AG08466), 2.1×10^6 A.U. (patient AG11513), 2.3×10^6 A.U. (patient AG02697), and 2.7×10^6 A.U. (patient AG06917).

7. (1) The authors do not show the increased formation of actin filaments, nor any of the other cellular phenotypes (except from decreased cilium length) in the different tissues of the *Lmna* null mice. (2) Do these mice not show these phenotypes? (3) And why is only cilium length indicated? Is cilium frequency not affected? This should be indicated.

Response:

(1) As suggested by the reviewer, we have tried to stain F-actin in the mouse tissues with phalloidin or anti-actin antibody and some basal body proteins, such as CP110 and Arp2, with specific antibodies. Unfortunately, none of these attempts succeed. The antibodies used in this study for immunofluorescence stain in cultured cells are not working for immunohistochemistry on mouse tissues.

(2) *Lmna*^{-/-} mice show reduced size in the tibialis anterior skeletal muscle, one of the characteristics of muscular dystrophy (Fig 2A). The altered cilia length is actually a phenotype that is most often described in the ciliopathy tissues (Pazour et al., 2000; Armour et al., 2012). We noticed at first glance that cilia are apparently shorter in the muscle and uterus of *Lmna*^{-/-} mice.

(3) As suggested by the reviewer, the cilia frequency was measured in mouse tissues. The results in new Fig 2F show that the cilia frequency is decreased in the muscle and uterus, but not in ovary and kidney.

8. The authors show that deficiency of lamin A in the mice cause cilium length defects, and relate that to a role of cilia in laminopathies. However, they don't show a cystic kidney phenotype in these mice, nor do they mention other hallmark ciliopathy phenotypes such as retinopathy or L/R randomization. Were these not observed? This should be evaluated and discussed.

Response:

Above mentioned ciliopathy phenotypes were not observed in *Lmna*^{-/-} mice, which may be due to the early death of *Lmna*^{-/-} mice. For example, polycystic kidney disease occurs at 30-40 years of age in humans. The medial life span of *Lmna*^{-/-} mice is only 42 days, which is

equivalent to 6 years of age in humans. Therefore, it is possible that *Lmna*^{-/-} mice die before the appearance of the ciliopathy phenotypes. In addition, we found that the adverse effect of lamin A deficiency on ciliogenesis is more apparent in muscle and uterus than in kidney and ovary (Fig. 2), suggesting that the significance of lamin A in ciliogenesis may be tissue-dependent. This may partially explain why lamin A deficiency does not cause same phenotypes as in ciliopathies.

9. (1) As the authors have access to *Lmna* null mice, why were these not deployed to evaluate the embryonic fibroblasts for the characteristics where they now use shRNA depletion instead? These are excellent cells for imaging and widely used to evaluate ciliopathy-associated mechanisms. Validating at least a subset of the cellular phenotypes in these cells would significantly strengthen the story, and (2) would also allow evaluation of signalling pathways such as Hedgehog signalling and mTor signalling, that are commonly affected in ciliopathies. (3) Even the effect of potential drugs such as rapamycin, which they mention in their discussion, could then be easily evaluated.

Response:

- (1) As suggested by the reviewer, we employed *Lmna*^{-/-} MEFs in this study. We found that *Lmna*^{-/-} MEFs displayed fewer and shorter primary cilia than *Lmna*^{+/+} MEFs, which was restored by cytochalasin D treatment (Appendix Fig S6). In addition, the deficiency of lamin A caused a decreased level of nesprine 2 and its abnormal cytoplasmic distribution (Appendix Fig. S9). The relative ratio of the nuclear G-actin in *Lmna*^{-/-} MEFs was lower than in *Lmna*^{+/+} MEFs (Fig EV3). Therefore, *Lmna*^{-/-} MEFs display defects similar to those of RPE cells induced by shRNA depletion.
- (2) As suggested by the reviewer, we evaluated the Sonic Hedgehog (Shh) signalling pathway in *Lmna*^{-/-} MEFs. Our preliminary results showed the Shh signaling appeared not affected in *Lmna*^{-/-} MEFs (as shown below). However, further studies in different cell types are necessary before we can draw a conclusion. In this study, we found that the adverse effect of lamin A deficiency on ciliogenesis is more apparent in muscle and uterus than in kidney and ovary of *Lmna*^{-/-} mice (Fig. 2). In addition, the motile the motile cilia lining the oviducts and bronchi appeared normally in *Lmna*^{-/-} mice (Appendix Fig S5). These results strongly suggest that the significance of lamin A in ciliogenesis is likely to be tissue-dependent. [Figures for referees not shown.]

(3) As suggested, the effect of rapamycin on ciliogenesis was examined. Our preliminary results showed that rapamycin at 5 nM or 1 μ M did not restore ciliogenesis in *Lmna*^{-/-} MEFs (data not shown). Further studies in different cell types are necessary before a conclusion can be drawn.

10. The only rationale for a relationship between nuclear lamina integrity and ciliogenesis is the G-actin experiment, although this relationship could be very well indirect: increased F-actin could trigger many defects in the cell, several of which could contribute to ciliogenesis defects, such as mislocalization of many actin binding proteins that are important for cilium function and ciliogenesis. Also in light of this, it is important to evaluate if the cell cycle is affected, and if the actin stress fibers/F-actin activation and cellular remodelling are also found in the patient cells and in the *Lmna* null mice .

Response:

- (1) The actin filaments were apparently increased in HGPS fibroblasts (Fig. 1D). The relative ratio of G-actin in the nucleus was decreased in both HGPS fibroblasts (Fig EV3A) and *Lmna*^{-/-} MEFs (Fig EV3B).
- (2) Our results show that the cell cycle of HGPS fibroblasts and *Lmna*^{-/-} MEFs was not apparently altered (as shown below).

[Figures for referees not shown.]

11. Arp2 localization is used as a measure for actin dynamics activity, but it seems to remain strictly localized to the centrioles. It is therefore important to also evaluate other components of this machinery (the Arp2/3 complex contains seven subunits of which Arp2 is just one, and actin dynamics involves many potential contributors) to validate these findings.

Response:

As suggested by the reviewer, we performed immunofluorescence staining to examine whether Arp3 and WASP are also localized at the basal body. Our preliminary results showed that Arp3 and WASP were localized at a region adjacent to the basal body in ciliated RPE cells (as shown below), which is not exactly the same as Arp2.

[Figures for referees not shown.]

12. The downregulation of nesprin 2 is a surprising finding and was suggested to be related to lamin A/C function mechanistically. Therefore, it is important to show (validate) and discuss

if this downregulation is consistent in the patient cells and *Lmna* null mice. Also, Dawe et al. (Ref. 38) showed that Nesprin 2 associates with Meckelin, a ciliopathy-associated protein, and that depletion of Meckelin leads to reduction of Nesprin 2, similar to lamin A/C. Interestingly, this correlated with an upregulation of activated RhoA, suggesting regulation via the ROCK-RhoA signalling pathway. In light of these earlier studies, it would be very informative to evaluate a potential activation of RhoA also in the lamin A depleted cells, and/or a potential co-depletion of meckelin.

Response:

- (1) The downregulation of nesprin 2 was observed in RPE cells with shRNA to lamin A (Fig 5A) and in *Lmna*^{-/-} MEFs (Appendix Fig S9). Unfortunately, the anti-nesprin 2 antibody available in our laboratory did not work for immunohistochemistry on mouse tissues.
- (2) Our preliminary results showed that RhoA appeared not activated in RPE cells with depletion of lamin A (data not shown).

Referee #2:

1. Cilia quantification is most convincing in studies that use dual labeling with two cilia markers for conclusive identification of cilia. This could be through use of an additional axonemal marker e.g. an antibody against Arl13b or a basal body antibody e.g. against pericentrin or gamma tubulin (in addition to detection of acetylated tubulin). If dual labelling is not possible quantification in alteration of cilia incidence length could still be undertaken with a different antibody to further confirm results. I do not feel this is necessary for all experiments where cilia are quantified, but would be a good additional control for initial experiment in the study (such as those in Figure 1-3).

Response:

As suggested by the reviewer, we have confirmed the ciliary defects with dual labeling of Arl13b and γ tubulin in HGPS fibroblasts (Appendix Fig S3), the tissues of *Lmna*^{-/-} mice (Appendix Fig S4) and RPE cells with shLamin A (Appendix Fig S8).

2. The authors conclude the reduction in cilia incidence and length is a consequence of impaired ciliogenesis. They have induced ciliogenesis through serum starvation for 32 hours. This is a relatively long time that would be predicted to put all cells are in cultures into

stationary phase. In addition to ciliogenesis, changes in cilia dynamics may be a consequence of altered cilia resorption. The authors have not demonstrated conclusively that the changes they observe are solely a consequence of altered cilia formation. While this may be challenging to fully test, the conclusion would be strengthened by inclusion of data quantifying cilia formation over time after serum withdrawal (e.g. 2, 4, 8, 12, 16, 24, 32 hours).

Response:

As suggested by the reviewer, we performed time-course experiments and found that the deficiency of lamin A/C in RPE cells led to attenuated cilia formation from 24 h after serum withdrawal (new Fig. 3E).

3. In the experiments quantifying cilia incidence and length in HGPS fibroblast, shown in Figure 1, cells are classified as having high or low progerin expression. There is a big variability of progerin levels in the high progerin cells. Is the relationship between progerin levels and cilia length a linear inverse correlation across this range of progerin levels? This should be tested for the HGPS cell lines i.e. progerin levels vs cilia length should be plotted.

Response:

The results in Fig EV1 show an inverse correlation between cilia length and progerin level in HGPS fibroblasts with progerin intensity above 3.0×10^6 A.U. (patient AG01972), 3.9×10^6 A.U. (patient AG08466), 2.1×10^6 A.U. (patient AG11513), 2.3×10^6 A.U. (patient AG02697), and 2.7×10^6 A.U. (patient AG06917).

4. (1) For the quantification of actin filaments within $60 \mu\text{m}^2$ of the basal body, how was the basal body position identified. No labelling for basal body is shown in the immunofluorescent labelling presented. Position of the basal body should be based on this labelling. (2) It would also be useful to show on the representative images where the $60 \mu\text{m}^2$ area, in which actin filaments were quantified, is. Presumably this was a circle of $\sim 4.37 \mu\text{m}^2$ around the basal body?

Response:

(1) As suggested by the reviewer, pericentrin was labeled to mark the position of the basal body in the figures (new Fig 4D, 5C and 6G). In Fig 1D, the HGPS fibroblasts were already labeled with anti-progerin and anti-acetylated tubulin, which prevented the application of anti-pericentrin.

(2) The representative images in new Fig 1D, Fig 4D, 5C and 6G where the 60 μm^2 area, in which actin filaments were quantified, were marked with a circle.

5. The authors conclude disruption of F actin by cytochalasin D restores cilia assemble and length in high progerin cells. They do not present any images or quantification showing that the cytochalasin D treatment used reduced actin filaments around the basal body. This should be included.

Response:

The cytochalasin-induced disruption of F-actin around the basal body is shown in new Fig 4D.

6. In the *Lmna* null mouse tissues was there any change in primary cilia incidence compared to controls. The cilia length data is clear, but no comment is made on incidence.

Response:

As suggested by the reviewer, we measured the incidence of primary cilia in the tissues of *Lmna* null mice and found that the incidence of primary cilia in the muscle and uterus of *Lmna*^{-/-} mice was lower than the control mice (new Fig 2F).

7. The representative image presented for rescue of cilia in cells 're-expressing' lamin A does not support the quantitative data very well. Only 2 adjacent cells, of approximately 13, appear to have 'normal' cilia.

Response:

As suggested by the reviewer, the representative image was replaced by a new one to better support the quantitative data.

8. Figure 4. Parts d & e; similar to point 4 above, this data quantifies stress fibers in proximity to the basal bodies, yet basal bodies are not directly labelled. Antibody combinations that make this possible exist and it should be done.

Response:

As suggested by the reviewer, pericentrin was labeled to mark the position of the basal body in the figures (new Fig 4D, 5C and 6G) and the areas where actin filaments were quantified, were marked with a circle.

9. For the nesp1 knockdown experiment in Figure 5, again stress fibers around the basal body are quantified in cells where the basal body has not been immunolabelled.

Response:

In new Fig 5C, the basal body was labeled with anti-pericentrin.

10. The data on redistribution of G actin from the nucleus to the cytoplasm in lamin deficient cells is particularly interesting as it provides mechanistic insight into the observations of the MS. Is a redistribution of actin from the nucleus also observed in the HGPS fibroblasts and does this correlate with progerin levels? This should be tested. Can the re localization of actin from the nucleus to the cytoplasm be quantified by any other means to support the fluorescence intensity data? Subcellular fractionation followed by immunoblot or FRAP experiments would be options.

Response:

- (1) The relative distribution of G-actin in the nucleus of HGPS fibroblasts was lower than that of normal fibroblasts (Fig EV3), however, which did not seem to correlate with the progerin level.
- (2) As suggested the reviewer, we performed subcellular fractionation and found that the relative level of actin in the nuclear fraction was decreased by deletion of lamin A/C (Appendix Fig S10).

11. For figure 7 it would be more convincing if further examples of Arp2 localization were shown. Can low power images with multiple centrosomes/cilia be shown with zoomed panels for the centrioles. This is an important finding, but showing a single representative image and the quantification is not ideal. If there are space constraints in the MS this could be included in the supplemental.

Response:

As suggested by the reviewer, we provide low power images with multiple centrosomes/cilia with zoomed panels for the centrioles (Appendix Fig S11).

12. (1) In the results (page 8) the MS states that '... lamin A/C depletion caused ~30% decrease in the number of basal bodies with TTBK2-recruitment and CP-110 removal'. Is dual labelling possible to show that the basal bodies with increased TTBK2 are more likely to lose

CP110? (2) These results are presented and discussed in term of recruitment of TTBK2 and CP110 to cilia, immunoblots to show total cellular levels of these proteins are not changed by depletion of lamin should be included.

Response:

- (1) The previous study has shown that TTBK2 is required for CP110-removal (Goetz et al., 2012). Because the antibodies used in this study for labeling TTBK2 and CP110 are rabbit polyclonal antibodies, dual labeling of both proteins is not applicable.
- (2) The expression levels of TTBK2 and CP110 are not affected by depletion of lamin A/C in RPE cells (Appendix Fig S12)

13. In the discussion the authors suggest that defective cilia function may be critical to the etiology of laminopathies. This argument would be strengthened if primary cilia function, rather than just structure, was analyzed in the lamin depleted cells. Shh signaling and other reporter assays are available to test this.

Response:

As suggested by the reviewer, we evaluated the Sonic Hedgehog (Shh) signalling pathway in *Lmna*^{-/-} MEF. Our preliminary results showed the Shh signaling appeared not affected in *Lmna*^{-/-} MEF (as shown below). However, more studies in different cell types are necessary before we can draw a conclusion. In this study, we found that the adverse effect of lamin A deficiency on ciliogenesis is more apparent in muscle and uterus than in kidney and ovary of *Lmna*^{-/-} mice (Fig. 2). In addition, the motile the motile cilia lining the oviducts and bronchi appeared normally in *Lmna*^{-/-} mice (Appendix Fig S5). These results strongly suggest that the significance of lamin A in ciliogenesis is likely to be tissue-dependent.

[Figures for referees not shown.]

Minor comments

1. Labelling of lanes for the immunoblot presented in Figure 3a could be improved. Shorten control to 'Con' and or angle at 45o. Or even better show if each treatment/condition is +/- for each lane of the blot.
2. There are a number of badly constructed sentences and grammatical errors in the MS that need to be addressed.
3. The introduction and discussion could be more concise including a less speculative discussion.

Response:

The manuscript has modified as the comments.

References

- Armour, E.A., R.P. Carson, and K.C. Ess. 2012. Cystogenesis and elongated primary cilia in Tsc1-deficient distal convoluted tubules. *Am. J. Physiol. Renal. Physiol.* 303:584-92
- Capell, B.C., M.R. Erdos, J.P. Madigan, J.J. Fiordalisi, R. Varga, K.N. Conneely, L.B. Gordon, C.J. Der, A.D. Cox, and F.S. Collins. 2005. Inhibiting farnesylation of progerin prevents the characteristic nuclear blebbing of Hutchinson-Gilford progeria syndrome. *Proc. Natl. Acad. Sci. USA.* 102:12879-84.
- Cajane, L. and E.A. Nigg. 2014. Cep164 triggers ciliogenesis by recruiting Tau tubulin kinase 2 to the mother centriole. *Proc. Natl. Acad. Sci. USA.* 28:2841-2850.
- Chen, Z.J., W.P. Wang, Y.C. Chen, J.Y. Wang, W.H. Lin, L.A. Tai, G.G. Liou, C.S. Yang, and Y.H. Chi. 2014. Dysregulated interactions between lamin A and SUN1 induce abnormalities in the nuclear envelope and endoplasmic reticulum in progeric laminopathies. *J. Cell Sci.* 127:1792-1804
- Chen, C.Y., Y.H. Chi, R.A. Mutalif, M.F. Starost, T.G. Myers, S.A. Anderson, C.L. Stewart, and K.T. Jeang. 2012. Accumulation of the inner nuclear envelope protein Sun1 is pathogenic in progeric and dystrophic laminopathies. *Cell.* 149:565-77
- Eriksson, M., W.T. Brown, L.B. Gordon, M.W. Glynn, J. Singer, L. Scott, M.R. Erdos, C.M. Robbins, T.Y. Moses, P. Berglund, A. Dutra, E. Pak, S. Durkin, A.B. Csoka, M. Boehnke, T.W. Glover, and F.S. Collins. 2003. Recurrent de novo point mutations in lamin A cause Hutchinson-Gilford progeria syndrome. *Nature.* 423(6937):293-8
- Goetz, S.C., K.F. Liem, Jr., and K.V. Anderson. 2012. The spinocerebellar ataxia-associated gene Tau tubulin kinase 2 controls the initiation of ciliogenesis. *Cell.* 151:847-858.

- Goldman, R.D., D.K. Shumaker, M.R. Erdos, M. Eriksson, A. E. Goldman, L.B. Gordon, Y. Gruenbaum, S. Khuon, M. Mendez, R. Varga, and F.S. Collins. 2004. Accumulation of mutant lamin A causes progressive changes in nuclear architecture in Hutchinson–Gilford progeria syndrome. *Proc. Natl. Acad. Sci. USA*. 24:8963–8968
- Pazour, G.J., B.L. Dickert, Y. Vucica, E.S. Seeley, J.L. Rosenbaum, G.B. Witman, and D.G. Cole. 2000. Chlamydomonas IFT88 and its mouse homologue, polycystic kidney disease gene *tg737*, are required for assembly of cilia and flagella. *J. Cell Biol.* 151:709-18.
- Schmidt, K.N., S. Kuhns, A. Neuner, B. Hub, H. Zentgraf, and G. Pereira. 2012. Cep164 mediates vesicular docking to the mother centriole during early steps of ciliogenesis. *J. Cell Biol.* 199:1083-1101.
- Wu, C.T., H.Y. Chen, and T.K. Tang. 2018. Myosin-Va is required for preciliary vesicle transportation to the mother centriole during ciliogenesis. *Nat. Cell Biol.* 20:175-185.

Dear Prof. Chen

Thank you for the submission of your revised manuscript to EMBO reports. We have now received the full set of referee reports that is copied below.

As you will see, both referees are very positive about the study and support publication after some remaining issues have been addressed. Referee 1 suggests performing a phalloidin staining to analyse whether there is an increased formation of actin filaments. Moreover, the referee recommends including important new data currently shown only in the response to the referees in the article and I agree with this suggestion.

From the editorial side, there are also a few things that we need before we can proceed with the official acceptance of your study.

1) Please note that a "Data availability" section at the end of Materials and Methods is mandatory. You can state that you have not generated data that require deposition in a public database in this section.

2) Please correct the header of the Methods section to 'Materials and Methods'

3) Please note that EMBO Reports will change from the current numbered reference style to the Harvard style as of July 1st (date of publication). I therefore kindly ask you to update the reference style accordingly. The respective EndNote file is available here
https://endnote.com/style_download/embo-reports/

4) Appendix:

a) please provide page numbers and list them in the Appendix table of content.

b) Please note that all material and methods must be part of the main manuscript unless they provide very specialized information, which is not the case here. Therefore, please remove the Appendix Supplementary Methods and add them to the methods section of the manuscript.

5) I notice that the Western blot in Figure 3A has been contrast modified to quite some extent and I suggest reducing the contrast settings so that the panel more reflects the result in the source data file.

6) Figure S2 seems to have rather low resolution (maybe a compression issue with the pdf file?) and this results in a much reduced Progerin staining (if compared to source data). Maybe the quality of the this figure can be improved?

7) Please add scale bars and define them in the legend for all magnification images (e.g, the insets in Figure 1A). This also applies to the Appendix insets.

8) Appendix Figure legends:

- Fig. S3C, D: please define the number of independent experiments in the legend and specify the nature of the bars in C and the error bars in D.

- Fig. S5: please define the white boxes

- Fig. S8B, C: Please define the number of replicates, the nature of the bars and error bars.

9) I attach to this email a related manuscript file with comments by our data editors. Please address all comments and upload a revised file with tracked changes with your final manuscript submission. I have also taken the liberty to make some changes to the Abstract. Could you please review it?

10) Finally, could you please provide a draft for the summary text that will be displayed next to the synopsis image? We require (A) a short (1-2 sentences) summary of the findings and their significance, and (B) 2-3 bullet points highlighting key results. Thank you.

Yours sincerely,

Martina Rembold, PhD
Editor
EMBO reports

Referee #1:

The authors have addressed most of my concerns in a thorough manner, and revised the manuscript, in most cases, appropriately.

One important concern I raised however (point 7 in the rebuttal) was that the authors omitted to show an increase in the formation of actin filaments in the tissues of the Lmna null mice. Their response, that staining of none of the proteins indicated in the mouse tissues are not working, is not satisfactory: This may be the case for specific antibodies, but phalloidin staining of mouse tissues is commonly rather trivial, and should provide a convincing answer to this important point.

The only remaining other comment I have, is that some of the issues I raised are addressed well, even experimentally with new Figures, but descriptions and Figures are only included in the point-by-point response and not in the manuscript itself. For several of these however, it is important to include the data and/or discussion thereof in the manuscript. This concerns the responses to:

(4) FACS analysis of RPE cells.

(5) Differential expression levels of progerin

(8) Lack of ciliopathy characteristics in Lmna^{-/-} mice

(9) - (2) Shh signalling assay

(10) - (2) Cell cycle evaluation in HGPS fibroblasts and Lmna^{-/-} MEFs

(11) Localization of Arp3 and WASP to the centrioles

Referee #2:

The authors have adequately addressed my initial questions and concerns regarding their manuscript with the inclusion of significant additional data and controls. This has strengthened the manuscript and the conclusion the authors make.

Referee #1:

1. One important concern I raised however (point 7 in the rebuttal) was that the authors omitted to show an increase in the formation of actin filaments in the tissues of the *Lmna* null mice. Their response, that staining of none of the proteins indicated in the mouse tissues are not working, is not satisfactory: This may be the case for specific antibodies, but phalloidin staining of mouse tissues is commonly rather trivial, and should provide a convincing answer to this important point.

Response:

In this study, tissues from *Lmna*^{+/+} and *Lmna*^{-/-} mice were embedded in paraffin. We searched articles and found that phalloidin staining is commonly employed in cryo-sectioned tissues, but rare in paraffin-embedded samples. However, we have made attempts to reveal F-actin in paraffin-embedded tissues from the control and *Lmna*^{-/-} mice. Unfortunately, no specific phalloidin signals were observed. We have tried several staining conditions, including 3 antigen retrieval methods, 2 phalloidin dyes, 4 phalloidin concentrations, and different staining temperatures/time (Table 1). The representative negative staining results are shown in Fig 1. A to E. We also sent our tissue samples to a biotech company (BioTools, Inc. Taiwan) for F-actin staining. Again, no specific phalloidin signals were revealed in the paraffin-embedded tissues (Fig 2. A to D). A possible explanation for the failure of phalloidin staining may be due to the loss of actin ultrastructure during the initial tissue processing steps, like paraformaldehyde-fixation and paraffin-embedding. Since we do not have cryo-sectioned tissues or live *Lmna*^{-/-} mice in hands, we are not able to further test phalloidin-staining in differently processed tissue samples.

Alternatively, we stained the F-actin in MEFs (*Lmna*^{+/+} vs *Lmna*^{-/-}) and demonstrated that lamin A depletion is associated with increased F-actin around cilia and increased cell spreading in the *Lmna*^{-/-} MEFs (Fig 3. A and B). These results may serve as supporting evidence for increased F-actin in *Lmna*^{-/-} mice. The data are included in Appendix Figure S8.

[Figures for referees not shown.]

2. The only remaining other comment I have, is that some of the issues I raised are addressed well, even experimentally with new Figures, but descriptions and Figures are only included in the point-by-point response and not in the manuscript itself. For several of these however, it is important to include the data and/or discussion thereof in the manuscript. This concerns the responses to:

- (4) FACS analysis of RPE cells. (5) Differential expression levels of progerin.
- (8) Lack of ciliopathy characteristics in *Lmna*^{-/-} mice. (9) Shh signalling assay.
- (10) Cell cycle evaluation in HGPS fibroblasts and *Lmna*^{-/-} MEFs.
- (11) Localization of Arp3 and WASP to the centrioles.

Response:

- a. In response to (4) and (10), the data of FACS analysis of RPE cells, HGPS fibroblasts and *Lmna*^{-/-} MEFs are included in Appendix Fig S12.
- b. In response to (9) and (11), the data are included in Appendix Fig S9 and Appendix Fig S14, respectively.
- c. In response to (5) and (8), our explanations are included to the first paragraph of the Results section and the fourth paragraph of the Discussion section, respectively.

Prof. Hong-Chen Chen
National Yang-Ming University
No. 155, Sec 2, Li-Nong St.
Taipei 11221
Taiwan

Dear Prof. Chen,

I am very pleased to accept your manuscript for publication in the next available issue of EMBO reports. Thank you for your contribution to our journal.

At the end of this email I include important information about how to proceed. Please ensure that you take the time to read the information and complete and return the necessary forms to allow us to publish your manuscript as quickly as possible.

As part of the EMBO publication's Transparent Editorial Process, EMBO reports publishes online a Review Process File to accompany accepted manuscripts. As you are aware, this File will be published in conjunction with your paper and will include the referee reports, your point-by-point response and all pertinent correspondence relating to the manuscript.

If you do NOT want this File to be published, please inform the editorial office within 2 days, if you have not done so already, otherwise the File will be published by default [contact: emboreports@embo.org]. If you do opt out, the Review Process File link will point to the following statement: "No Review Process File is available with this article, as the authors have chosen not to make the review process public in this case."

Should you be planning a Press Release on your article, please get in contact with emboreports@wiley.com as early as possible, in order to coordinate publication and release dates.

Thank you again for your contribution to EMBO reports and congratulations on a successful publication. Please consider us again in the future for your most exciting work.

Yours sincerely,

Martina Rembold, PhD
Editor
EMBO reports

THINGS TO DO NOW:

You will receive proofs by e-mail approximately 2-3 weeks after all relevant files have been sent to our Production Office; you should return your corrections within 2 days of receiving the proofs.

Please inform us if there is likely to be any difficulty in reaching you at the above address at that time. Failure to meet our deadlines may result in a delay of publication, or publication without your corrections.

All further communications concerning your paper should quote reference number EMBOR-2019-49680V3 and be addressed to emboreports@wiley.com.

Should you be planning a Press Release on your article, please get in contact with emboreports@wiley.com as early as possible, in order to coordinate publication and release dates.

PLEASE NOTE THAT THIS CHECKLIST WILL BE PUBLISHED ALONGSIDE YOUR MANUSCRIPT

Corresponding Author Name: Hong-Chen Chen

Manuscript Number: EMBOR-2019-49680-T